# communications
## earth & environment

# Influence of reef isostasy, dynamic topography, and glacial isostatic adjustment on sea-level records in Northeastern Australia

Alessio Rovere [1,2,10✉], Tamara Pico [3,10✉], Fred Richards [4], Michael J. O'Leary[5], Jerry X. Mitrovica[6], Ian D. Goodwin[7,8], Jacqueline Austermann[9] & Konstantin Latychev[6]

Understanding sea level during the peak of the Last Interglacial (125,000 yrs ago) is important for assessing future ice-sheet dynamics in response to climate change. The coasts and continental shelves of northeastern Australia (Queensland) preserve an extensive Last Interglacial record in the facies of coastal strandplains onland and fossil reefs offshore. However, there is a discrepancy, amounting to tens of meters, in the elevation of sea-level indicators between offshore and onshore sites. Here, we assess the influence of geophysical processes that may have changed the elevation of these sea-level indicators. We modeled sea-level change due to dynamic topography, glacial isostatic adjustment, and isostatic adjustment due to coral reef loading. We find that these processes caused relative sea-level changes on the order of, respectively, 10 m, 5 m, and 0.3 m. Of these geophysical processes, the dynamic topography predictions most closely match the tilting observed between onshore and offshore sea-level markers.

[1] Department of Environmental Sciences, Informatics and Statistics, Ca' Foscari University of Venice, Venice, Italy. [2] MARUM - Center for Marine Environmental Sciences, University of Bremen, Bremen, DE, Germany. [3] Earth & Planetary Sciences Department, UC Santa Cruz, Santa Cruz, CA, USA. [4] Department of Earth Science & Engineering, Imperial College London, London, UK. [5] School of Earth Sciences, University of Western Australia Oceans Institute, Perth, WA, Australia. [6] Department of Earth and Planetary Sciences, Harvard University, Boston, MA, USA. [7] Climalab, Sydney, NSW, Australia. [8] Climate Change Research Centre and Australian Centre for Excellence in Antarctic Science, University of New South Wales, Kensington, NSW, Australia. [9] Department of Earth and Environmental Sciences & Lamont-Doherty Earth Observatory, Columbia University, New York, NY, USA. [10] These authors contributed equally: Alessio Rovere, Tamara Pico. ✉email: alessio.rovere@unive.it; tpico@ucsc.edu

Below the modern Great Barrier Reef (GBR) reef flats, coring has typically encountered shallow-water Last Interglacial (LIG, MIS 5e, 125 kyrs) reefs between depths of 5 and 20 m. Strikingly, along the Queensland and far northern New South Wales coastline, LIG strandplains are identified at higher elevations than offshore LIG reefs, with ridge/swale heights ranging from +3 to +9 m above modern sea level.[1,2] These onshore features are not as precisely dated as the sea-level indicators found within fossil reefs in cores, however, they were also arguably formed during the LIG. The higher elevations of these coastal strandplains are roughly consistent with estimates for peak LIG global mean sea level (GMSL). Such estimates are consistently above modern mean sea level (0 m), albeit they vary substantially depending on study sites analyzed and corrections for vertical land motions applied to the proxy record (from 6 to 9 m[3], 8 m[4], and 1–5 m[5]).

The most obvious explanation of the discrepancy between onshore and offshore LIG relative sea-level indicators in Northeastern Australia is that these two areas are subject to differential vertical land motions. When reconstructing past GMSL from geological sea-level proxies, it is essential to disentangle the components causing globally averaged sea-level changes from other regional processes that may have caused vertical displacement of past sea-level indicators[6,7]. Among these, the most relevant are glacial isostatic adjustment (GIA)[8], tectonic deformation processes[9], and mantle dynamic topography (DT)[10].

Crustal loading due to local processes can also cause the vertical displacement of observed sea-level indicators through isostatic adjustment. For example, sediment loading can cause regional sea levels to depart significantly from the global mean along major deltaic systems[11–16]. Karst erosion is another mechanism that induces isostatic adjustment, through mass unloading, causing a net crustal uplift. This process is active in the Plio-Pleistocene shoreline complexes in Florida that were uplifted following isostatic response to the karstification (leading to rock mass loss) of the landscape[17–20]. To date, estimates of peak LIG GMSL from tropical areas have not accounted for the isostatic response to coral reef loading over the last glacial cycle. This process arises because corals can grow into spatially extensive reefs, reaching thicknesses of several tens of meters during interglacials. The effect of reef accretion and related loading on local sea-level histories remains largely unexplored.

In this work, we model the influence of geophysical processes that may have changed the elevation of geologic sea-level indicators along the Queensland coasts and offshore, on the GBR, since the LIG. We assess the extent to which the combined geophysical processes of GIA and DT may have impacted the LIG sea-level record in this region. We also isolate the process of coral reef loading, and assess its contribution to regional departures from GMSL. While the combined geophysical processes modeled in this study cannot fully explain the amplitude of the observed discrepancy between onshore and offshore sea-level markers in the study area, we find that dynamic topography contributes the largest magnitude to the observed tilting.

## LIG sea-level indicators

The study of past sea-level changes relies on the measurement and dating of relative sea-level (RSL) indicators, i.e. geological proxies that formed in connection with former positions of the sea. Once a sea-level indicator is measured and dated, it is necessary to establish its indicative meaning[21,22] to quantify the relationship between the elevation or depth of an indicator and the position of the former sea level, including associated uncertainties due to the environmental range of formation. The corrected elevation of a sea-level indicator reflects paleo RSL, i.e., the paleo position of the sea including both barystatic (i.e., eustatic,[23]) changes, elevation changes due to vertical land motions of different origin, and perturbations in the sea surface height.

On the GBR, corals of LIG age are presently preserved under a subsurface unconformity, which occurs down to 20–25 m below present sea level, depending on the site[1,24–26]. Murray-Wallace and Belperio[1] highlight that while low-lying islands are scattered throughout the GBR, outcrops of Pleistocene reefs above modern sea level are absent. The only exception may be an exposed reef of apparently Pleistocene age at 1–4 m above present sea level[24] at Digby Island[27,28]. However, the age of this reef has never been confirmed with absolute dating, and it will not be discussed further. Retrieval of LIG reef sections on the GBR has been historically done by coring through the Holocene reef down to the Holocene/LIG unconformity. A full account of the best-preserved and best-dated Last Interglacial corals on the GBR, alongside the paleo water depth of the coralgal assemblages and sedimentary facies associated with them, is provided by Dechnik et al.[29]. These data were recently compiled into the standardized WALIS (World Atlas of Last Interglacial Shorelines) database by Chutcharavan and Dutton[30] (blue markers in Fig. 1). In general, these reefs have paleo water depths <3 m or <6 m, therefore they developed in very shallow waters. The shallowest reef unit dated to MIS 5e (131 ± 1 ka, after open-system U-series corrections) was recently reported at Holbourne Island[26], at ca. 5 m below the Lowest Astronomical Tide. It is worth noting that this island is much closer to the shoreline (20 km vs >50 km) and is morphologically different from those reported by Dechnik et al.[29], as it is a continental high island rather than a low-lying coral island. This dated reef was not included among those reported in this work as we could not find enough information to produce a reliable sea-level index point from the information provided by Ryan et al.[26].

Murray-Wallace and Belperio[1] report the presence of scattered coastal deposits of LIG age along the continental coasts of New South Wales and Southern Queensland. These were interpreted, according to their sedimentary and geomorphological characteristics, as beach barriers, estuarine deposits, or dune-island barriers. These features are ubiquitous along the SE Queensland Fraser Island Coast and far north New South Wales coasts[2], where the LIG age of the deposits is confirmed by U-series on corals embedded in the sedimentary units or Amino Acid Racemization dates[1]. The LIG strandplains are often overlain by Holocene transgressive sequences. Similar deposits as those described in New South Wales and Southern Queensland are also present in our study area. However, in contrast to LIG reef sequences in the GBR, most of these strandplains are rarely assigned an age with absolute dating techniques. Their MIS 5e age has been inferred via analogy with the strandplains in New South Wales and Northern Queensland, chronostratigraphic correlation with lower younger (Holocene) units, and infinite radiocarbon ages. An expanding Optically Stimulated Luminescence chronology for these deposits is in progress[2], and shows that complete LIG strandplains are located inboard of the modern Holocene equivalents.

In far north Queensland, Gagan et al.[31] describe LIG dune/beach barrier located onshore with respect to the Holocene equivalent at Wyvuri Embayment (Fig. 2). According to Gagan et al.[31], the top of the barrier, composed of aeolian sediments, is located at +6 m above modern sea-level (in our topographic profile in Fig. 2 this plots slightly higher, 7.5 m), while the beach barrier sands were intercepted about 4 m below the surface, in drill cores. This elevation roughly corresponds to a break in slope on the coastal plain (3.4 ± 1.5 m), which can be interpreted as a shoreline angle. Considering this analog to a beach deposit, and using the formulas and values suggested by Lorscheid and Rovere[32] to calculate the indicative meaning in the absence of

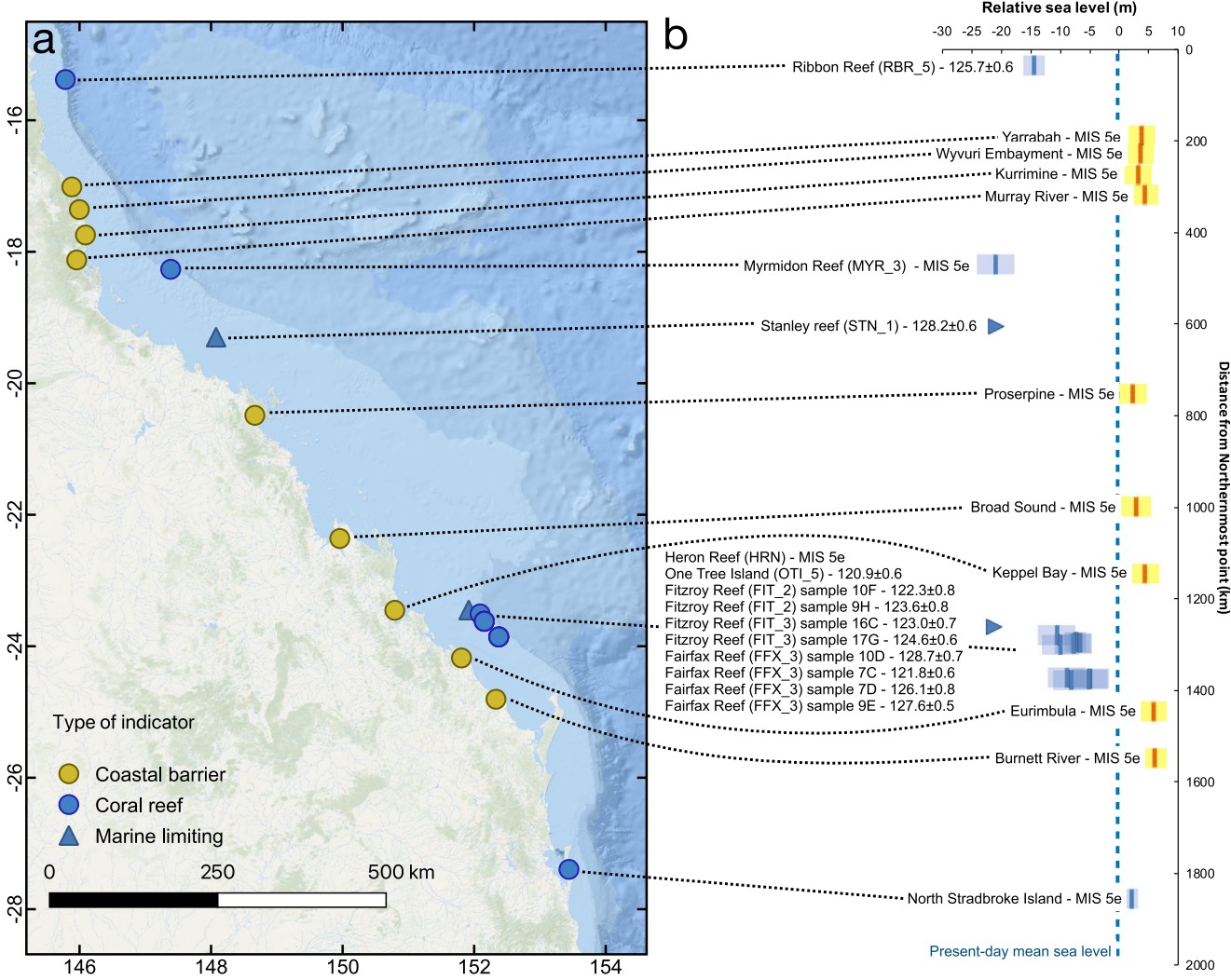

**Fig. 1 LIG sea-level index points and paleo RSL along the GBR and on the coasts of Northeastern Australia. a** map and **b** elevation plot of LIG paleo RSL obtained from fossil reefs (blue markers) and beach barriers (yellow markers) along the GBR and the Queensland Coasts. Error bars represent 1-sigma ranges. Basemap sources: Esri, GEBCO, NOAA, National Geographic, Garmin, HERE, Geonames.org, and other contributors.

modern analog data, we calculate that this strandplain indicates LIG paleo RSL of 3.4 ± 2.7 m (Fig. 2). At the nearby Cowley Beach strandplain, Brooke et al.[33] established that the strandplain beach ridge morphology tracked Holocene sea-level trends.

The surface expression of the Wyvuri Embayment LIG beach barrier can be found at other locations along the Queensland coast, with the shoreline angle located roughly at the same elevation as Wyvury Embayment (yellow markers in Fig. 1). Towards the south of our study area, near the border between Queensland and New South Wales, fossil corals embedded into beach/intertidal/shallow subtidal deposits at North Stradbroke Island, are overlain by Holocene transgressive deposits and were dated to MIS 5e[34,35]. The original authors suggest that these would indicate a paleo sea level between 1 and 3 m, which is consistent with the paleo sea level calculated from the other beach barriers described above.

Starting from the description of Gagan et al.[31] and high-resolution (5 m) Digital Elevation Models from ref. [36], we identified other locations scattered along the Queensland coast where the LIG has left a morphological imprint as an evident beach barrier on the strandplain, from which sea-level index points can be derived (see Supplementary Materials[37] for detailed maps of each area and a spreadsheet containing sea-level interpretations,

similar to those shown in Fig. 2). The elevation of these barriers is consistent with those identified in northern New South Wales, which preserve LIG sea-level trend from a highstand at +6 ± 0.5 m at 129 ka BP to +4 m by 116 ka[2]. The SE Queensland and northern New South Wales studies revealed that regional coastal fault re-activation has occurred during the Late Quaternary that has influenced the accommodation space for strandplain deposition. Overall the Late Quaternary onshore strandplains extending from far North Queensland to far northern New South Wales indicate that Late Pleistocene strandplains are preserved in the +3 to +6 m elevation. This is in stark contrast to the offshore submerged record, suggesting a LIG paleo relative sea level below the modern one.

The fact that LIG reefs in the GBR are found below the typical elevation of reefs of the same age on passive continental margins was discussed by Marshall and Davies[25], who attributed it to a combination of long-term subsidence of the continental margin and erosion of the Pleistocene reef framework during glacial times. Differential Holocene reef growth rates seem to indicate that the Central GBR is subsiding with respect to the Northern and Southern GBR. Dechnik et al.[38] suggest that this subsidence may be related to the re-activation of NNW-SSE extensional faults along the eastern Queensland margin[39], and references therein.

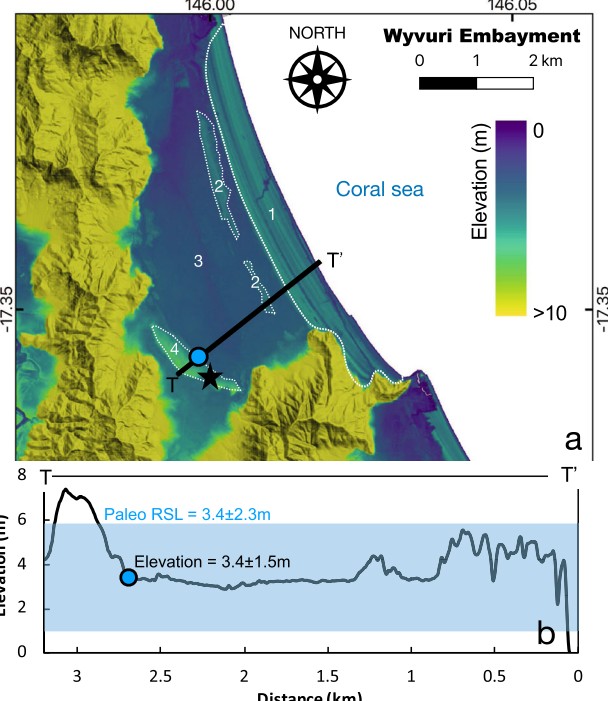

**Fig. 2 Example of LIG strandplain along the Queensland coast. a** Digital Elevation Model[36] and **b** topographic profile of the Wyvuri Embayment, where Gagan et al.[31] identified LIG coastal sediments in a core under a dune/beach barrier. The star indicates the approximate point where core JW4 of Gagan et al.[31] was drilled. Numbers 1-4 indicate the facies reported in Gagan et al.[31]: 1 Holocene beach barrier; 2 Holocene back-barrier; 3 Holocene freshwater swamp; 4 Last Interglacial beach barrier. The blue dot indicates the inner part of the LIG barrier used as a sea-level proxy in this study. The blue transparent overlay on the topographic profile indicates the paleo RSL calculated using the elevation of the inner margin of the barrier and the indicative meaning calculator tool[32].

## Results and discussion

**Reef isostasy.** Coral reefs are created by the fixation of calcium carbonate mostly by hermatypic corals and calcareous algae[40]. Reefs respond to variations in sea-level by catching up, keeping up, or giving up. From the geophysical perspective, this results in the creation of a mass of reef framework, which can exert a relevant load on the underlying crust. This loading causes an isostatic response that is non-negligible. Hereafter, we define the isostatic adjustment induced by coral reef building as "*reef isostasy*".

An illustration of how reef isostasy impacts the elevation of LIG reef measured today is shown in Fig. 3. During the LIG, a reef builds on top of an older reef surface (or the basement, Fig. 3a). This loading induces isostatic adjustment, causing subsidence, or equivalently a relative sea-level rise. The sea-level change ΔRSL magnitude induced by reef isostasy depends on reef thickness as well as its geographic extent. Areas with loads of smaller spatial scale are compensated more by elastic stresses, resulting in a smaller magnitude relative sea level change associated with reef isostasy. During a subsequent glacial period of lower sea level, erosion, and karstification may lead to unloading-induced uplift that partially compensates for the subsidence during reef-building (Fig. 3b). However, we do not model this process in this work, as the total mass change since the Last Interglacial is dominated by reef growth, rather than reef erosion.

An increase in local relative sea-level from crustal subsidence induced by reef isostasy results in lower elevation LIG coral

sea-level markers today, (assuming no GMSL difference) compared to their original elevation at the LIG. Therefore LIG coral reef sea-level marker elevations must be corrected upwards to account for reef isostasy, potentially resulting in higher reconstructed LIG GMSL than prior estimates.

**Modeling reef isostasy: fine vs. coarse resolution.** The predicted magnitude of relative sea level change is sensitive to the spatial scale of the load, in addition to the load thickness. We first perform calculations using a 3D sea-level model, and the "fine resolution grid" coral reef loading scenario with a regional spatial resolution of 1 km that accounts for the fractional area of reef coverage in each grid cell (Methods). We next compute reef isostasy using the "coarse resolution grid" to assess whether the lower resolution input accurately captures the crustal deformation (and thus relative sea level) response to reef loading. Note that these coarse resolution runs use a 1D GIA model setup and a loading scenario that does not account for reef coverage area resulting in a larger volume and mass load for the coarse resolution case (Methods).

Figure 4 (right panels) shows the elevation change that LIG sea-level indicator would undergo from 122 to 0 ka due to reef isostasy (negative values signify that sea-level indicators experienced subsidence since the LIG). Our fine resolution simulation of reef isostasy in the Great Barrier Reef predicts a maximum relative sea level change of 0.34 m since the Last Interglacial (Fig. 4b). These maximum values are reached in Northeastern Queensland and along the coastline of the southern GBR. Our predictions for relative sea level change due to reef isostasy suggest this process is small compared to other uncertainties on the paleoelevation of LIG coral reefs (for example, coral growth depths, tides etc.). In contrast, the coarse resolution reef isostasy calculations predict a maximum relative sea level change of 1.45 m since the Last Interglacial (Fig. 4d). The discrepancy between fine vs. coarse resolution models is due to the fact that the fine resolution calculation involves a more localized loading geometry (and thus reduced crustal deflection) due to elastic compensation within the lithosphere, compared with the coarse resolution case that overestimates the mass load by not accounting for areal extent on a finer resolution grid.

Because fine-resolution modeling using the 3D sea-level model is computationally expensive, we also tested whether a 1D sea-level model could accurately capture the pattern and magnitude of relative sea-level change due to reef isostasy. We first used the fine-resolution coral reef loading scenario and multiplied the loading grid by the fractional area of reef coverage on a 1 km-scale. We then interpolated this loading scenario onto a grid with ~34 km resolution to create a coarse grid that accounts for a fractional area of reef coverage (Fig. 4e). We ran a 1D sea-level model with this loading scenario using the same Earth model as in the other 1D calculation. This simulation resulted in a similar magnitude of reef isostasy as in the 3D fine resolution model, with a maximum value of 0.4 m of RSL change since the LIG (Fig. 4f). However, the spatial pattern does not reproduce the signal along the southern Great Barrier Reef coastline shown in the 3D fine resolution simulations. This difference is likely due to the higher resolution associated with the 3D sea-level simulation rather than 3D earth structure, as the coarse resolution 1D calculation does not capture the reef loading regions along the central and southern Great Barrier Reef coastline.

To assess the sensitivity of our results to Earth structure parameters, we also performed 1D sea-level simulations using an alternate Earth model, VM2[41]. We found that changing the Earth model had a negligible effect, perturbing the predicted RSL

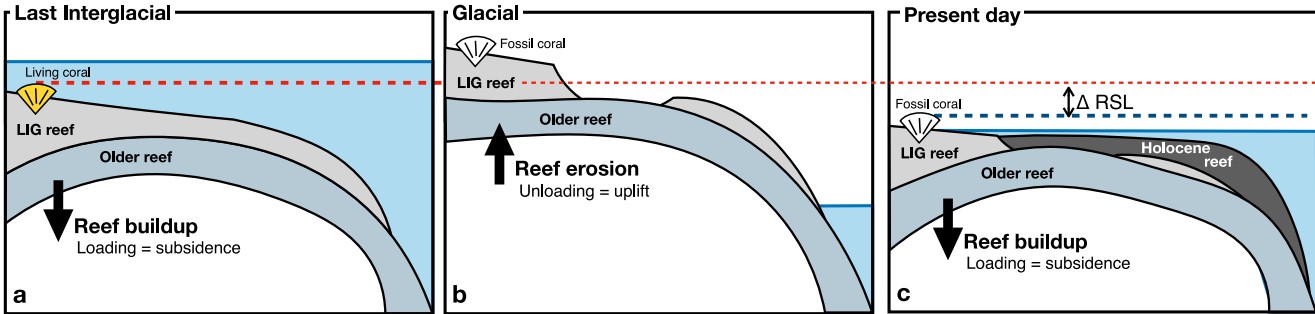

**Fig. 3 Illustration of reef isostasy caused by the buildup of the reef complex since the Last Interglacial. a** The LIG reef is built on top of an older reef (or the bedrock). The addition of this load leads to isostatic subsidence of the underlying bedrock. **b** As GMSL falls (e.g., under glacial conditions), the reef is partially eroded and/or dissolved (e.g., by karst processes), resulting in isostatic rebound. **c** As sea level rises a second time, the reef starts to build again on top of previous structures, causing additional subsidence. ΔRSL represents the relative sea-level change caused by reef isostasy. The colored dashed lines represent the elevation of the coral during the LIG (red) and its present-day elevation (blue). Note that the uplift and subsidence following reef loading and unloading are transient through glacial-interglacial times, and that in our study we do not model the uplift following reef erosion, which we consider to be balanced with Holocene re-growth.

change by a maximum of 3% at the Queensland/GBR sea-level indicator sites.

**Contribution of GIA and dynamic topography**. We predicted the elevation change due to reef isostasy (Fig. 5a), dynamic topography (Fig. 5b), and GIA (Fig. 5c) from 127 ka to present (see Methods for details). These values represent the elevation change a LIG sea-level indicator would undergo from 127 to 0 ka (negative values signify that sea-level indicators experienced subsidence, and positive values signify that sea-level indicators experienced uplift since the LIG). The total predicted influence on the Last Interglacial sea-level indicator elevation from these geodynamic processes is shown in Fig. 5d.

Our dynamic topography predictions show an elevation change of −10 to 10 m from 127 ka to present day, a rate of differential vertical motion that exceeds some regional estimates[42], but is comparable to others[43]. This means that dynamic topography would have uplifted the Australian continent by up to 10 m, while offshore regions on the continental shelf would have subsided up to 5 to 10 m since the LIG. Variations in input density and viscosity structure lead to ~±1 m uncertainty in post-LIG dynamic topography change (based on standard deviation of 15 model predictions), and the spatial pattern is remarkably consistent amongst the 15 models investigated here. These results suggest that our predictions of convectively driven onshore-offshore tilting are robust. This inference is corroborated by ~100 m Myr$^{-1}$ uplift rates inferred from river profile modeling[44] and patterns of Late Cenozoic age-independent magmatism[45], both features that have been attributed to the presence of an active small-scale convection cell beneath the Queensland margin. Although the dynamic topography maxima and minima are offset with respect to the observed relative sea level maxima and minima, the highest horizontal resolution for the dynamic topography predictions is ~200 km, and therefore it may not be possible to precisely match the observed tilting at this resolution.

Similarly, glacial isostatic adjustment would have produced uplift on the continent and subsidence offshore. Our predictions show that the continent may have uplifted 6 m and offshore regions subsided 2 m since the Last Interglacial. The spatial variability in elevation change due to glacial isostatic adjustment is caused by the process known as continental levering, where uplift occurs along continental margins as sea-level rise causes subsidence in ocean basins due to water loading[46,47].

In this study, we did not model several other potential mechanisms that may cause departure from eustasy in the study area. For example, crustal deformation due to re-activation of older faults has been inferred to affect Holocene reefs see ref. [39], and references therein. While such a mechanism might have a relevant local effect, any fault system causing crustal motions would have to be active (with roughly the same deformation rates) over ~2000 km of coast to reconcile the observed onshore-offshore tilting trend. This seems an unlikely pattern in an intraplate margin setting such as the Queensland-GBR area. Another process we did not model is erosion and sediment deposition which drive a tilting (up on land) of the crust. Studies on the Central GBR shelf suggested that the thickness of Holocene sediments is rather limited <2.5 m[48] hence siliciclastic sediment isostasy seems an unlikely explanation for the large difference between onshore and offshore LIG sea-level proxies, recorded over such a large latitudinal gradient.

An important caveat to our reef isostasy modeling is that we did not account for additional loading associated with other processes, such as carbonate sands (also mixed with siliciclastic sediments) close to modern reef areas[49,50], post-LGM reef buildups (now drowned on the shelf[49]), and other bioherms of considerable importance, such as inter-reefal *Halimeda* algal buildups[51]. Including these factors would increase the load and hence the relative importance of reef isostasy, however, it is unlikely to explain the large differences between the onshore and offshore LIG sea-level indicators.

## Conclusions
The Queensland-GBR area is characterized by an enigmatic difference in the elevation of LIG sea-level indicators between off-shore (GBR) and onshore (Queensland coast) sites. This offset motivated our modeling of local post-depositional vertical land motion. We modeled sea-level change due to reef isostasy, dynamic topography, and GIA since the LIG in this area, is located on a passive margin spanning a latitudinal range of almost 2000 km. Our models explored whether reef isostasy, which is considered here for the first time, may play a role in the observed vertical displacement of LIG fossil reefs, which are among the most frequently used geological sea-level proxies[52–54].

Our results show that the contribution of reef isostasy to vertical land motions is negligible, reaching maximum values of 0.34 m. In comparison with GMSL changes, this is roughly equivalent to half the contribution to GMSL of mountain glaciers melting and thermal expansion during the LIG (estimated as up to 1 m[55]). Reef isostasy, therefore, produces a relatively small change in RSL since the LIG at the GBR, and is insufficient in

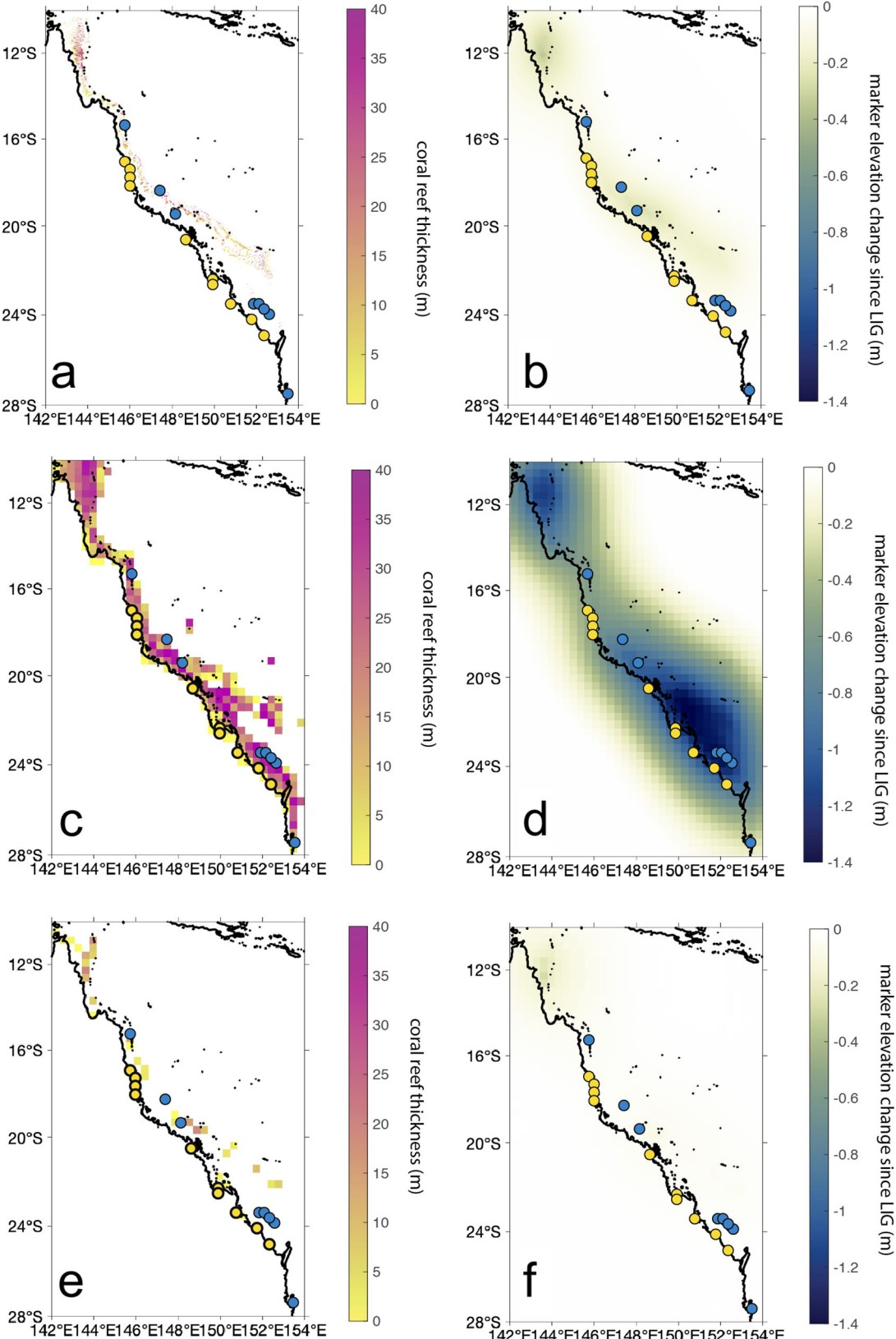

**Fig. 4 Reef thickness and reef isostatic response. a** Fine resolution coral reef thickness (122-0 ka) for the reef isostasy loading scenario. **b** Predicted marker elevation change since LIG due to reef isostasy in response to loading in frame **a**. **c**, **d** As in **a**, **b**, except for the coarse resolution modeling. **e**, **f** As in **c**, **d** except for the coarse resolution treatment of reef thickness (122-0 ka) accounting for reef area coverage. Yellow and blue dots on each map represent the sites shown in Fig. 1.

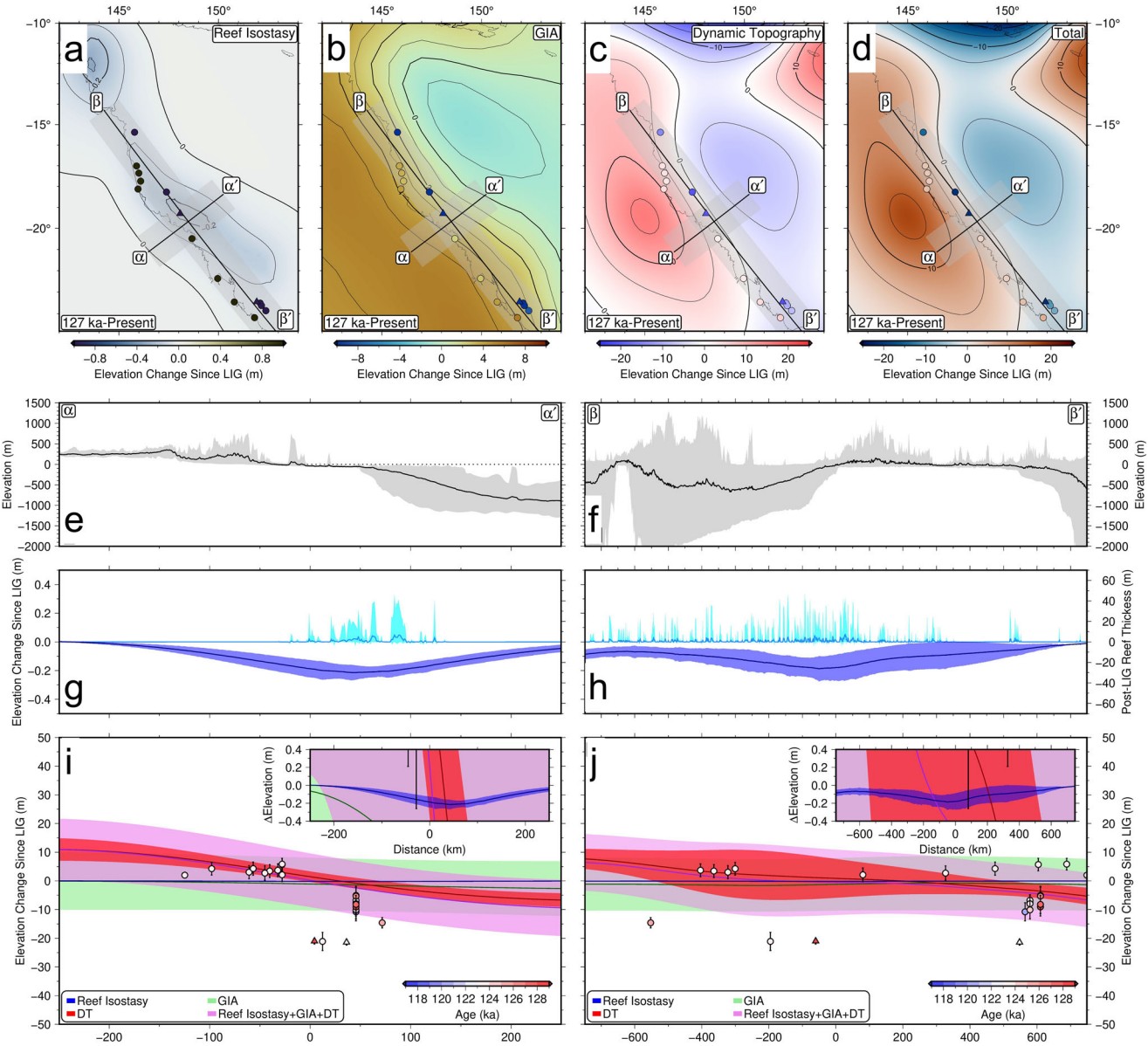

**Fig. 5 Summary of departures from eustasy in the study area.** Predicted elevation change to sea-level indicators from 127 to 0 ka due to: **a** reef isostasy, **b** glacial isostatic adjustment **c** dynamic topography. Colored circles represent LIG sea-level indicators as shown in Fig. 1. **d** Total predicted elevation change to sea-level indicators from 127 to 0 ka. **e**, **f** gray represents observed elevation range and black line represents mean values for transect α–α′ (left) and β–β′ (right). **g**, **h** Light blue line and envelope represents the observed range in reef thicknesses in coral reef loading scenario from LIG to present. Dark blue line and envelope represent the predicted elevation change to sea-level markers due to reef isostasy (as in **a**). Lines represent mean values based on spatial uncertainty of 100 km on either side of transect and intermodel variation uncertainty; envelopes represent the 2 sigma combined uncertainty. **i**, **j** GBR LIG sea-level data points projected onto transects α–α′ (**i**) and β–β′ (**j**) as a function of distance between the data point and the closest point on the transect. Colored circles/triangles represent LIG sea-level indicator ages. Predicted elevation change projected onto transect α–α′ (**i**) and β–β′ (**j**) for reef isostasy (blue), dynamic topography (red), glacial isostatic adjustment (green), and total (pink). Lines and envelope calculated as in **g**, **h**.

magnitude to explain discrepancies between observed LIG RSL markers offshore and onshore. However, we emphasize that the load we constructed might be an underestimation, so this mechanism may represent a potentially important contribution to vertical land motions in areas with dense and widespread coral reef coverage. Therefore, neglecting reef isostasy may represent a potential bias in areas with significant reef coverage.

To realistically represent coral reef loading since the LIG in a given area, it is important to gather direct measurements of reef thickness, extent, density, and porosity, together with estimates of mass loss since the LIG (e.g., due to erosion or karst processes,

which we do not model here) and, in the case of wide lagoons, carbonate sediment production rates from the reef. In addition, the presence of buildups other than coral reefs, capable of producing relevant loads at wide spatial scales, is important. Our results underscore the importance of fine resolution modeling, especially in accounting for the areal coverage of coral reefs, to accurately reproduce relative sea level change due to reef isostasy. Once these data are available, we show that while 1D sea-level models are more computationally efficient, for small-scale loading patterns such as coral reefs, it may be important to use high-resolution 3D modeling to accurately capture the relative sea level response to reef loading.

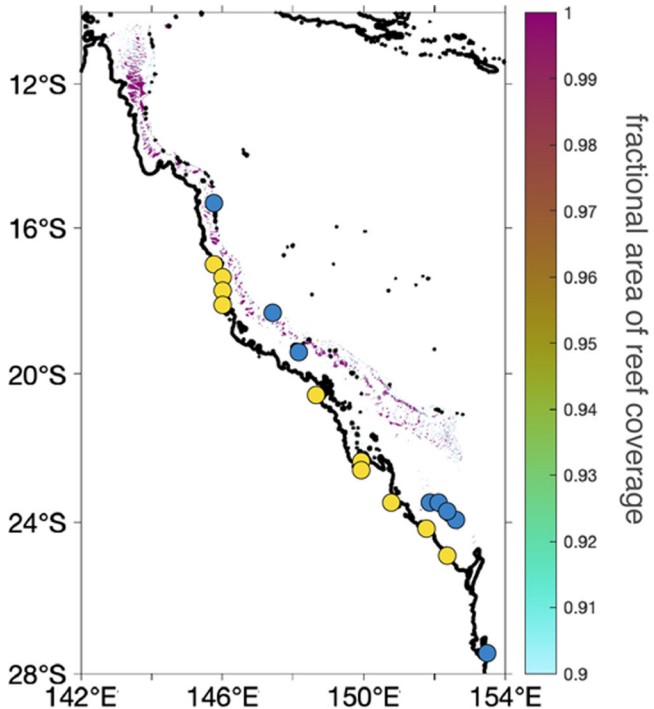

**Fig. 6 Fractional area of present-day reef coverage.** Yellow and blue dots represent the sites shown in Fig. 1.

Comparing the modeled relative contributions of reef isostasy, dynamic topography, and GIA, we surmise that only the predicted change due to dynamic topography across sites has a magnitude similar to the differences in sea-level indicator elevations between onshore and offshore. This result strengthens the argument that dynamic topography may play a major role in the vertical displacement of LIG sea-level indicators at Late Pleistocene time scales[10], and cannot be ignored, even at passive margins, in MIS 5e sea-level reconstructions.

## Methods

**Constructing the coral reef loading scenario.** As a baseline dataset for the presence/absence of coral reefs, we used the 500 × 500 m raster dataset[56–58] of the warm-water reefs map compiled by UNEP-WCMC, WorldFish Centre, WRI, TNC[59–62]. We created a coral reef loading scenario since the LIG (122-0 ka) using two methods, with different resolutions. For the "coarse resolution grid", we used a standard approach for sea-level model calculations and placed our coral loading scenario onto a ~ 34 km resolution grid. For the "fine resolution grid", we placed our coral loading scenario onto a 1 km resolution grid, and accounted for the areal fraction of coral reef coverage within each 1 × 1 km grid cell.

Because the GBR reef is characterized by narrow, sometimes isolated, strips of coral reef, we were concerned that the standard grid resolution (~34 km) used in sea-level models may unrealistically smooth out the reef loading signal. Thus, for the "fine resolution grid" we interpolated a high-resolution Digital Elevation Model for bathymetry in the Great Barrier Reef area onto a 1 km resolution grid[63]. We then assessed the fractional area of reef coverage within each 1 × 1 km grid cell using the "Fishnet" tool of ArcGIS. Of grid cells with non-zero reef coverage, 44% had full reef coverage (Fig. 6). We then multiplied the coral reef thickness in our 1 × 1 km grids by the areal fraction of reef coverage to produce our "fine resolution grid" coral reef loading scenario.

We also used a standard approach for constructing a loading scenario by interpolating a high-resolution bathymetric Digital Elevation Model of the GBR area onto a Gauss Legendre grid with ~34 km resolution (maximum spherical harmonic degree 512) commonly used in sea-level calculations. This approach does not account for coral reef coverage since the coral reef thickness is smoothed over a wide area relative to the lateral extent of coral reefs. We term this coral reef loading scenario the "coarse resolution grid" (Fig. 4c).

Apart from a very small number of examples, including the Ribbon Reef Core in the Northern GBR outer shelf (155 m reefal thickness), Boulder Reef core northern GBR mid shelf (33 m reeflal thickness)[64], and One Tree Reef core Southern GBR mid shelf (18 m reefal thickness)[38], the total vertical extent of reef buildups since the LIG is largely unknown. Limited seismic stratigraphy of the GBR has focused on the inter-reefal shelf areas and show the shelf comprising Permo-Carboniferous bedrock, Pleistocene/Tertiary sediments, consisting of both shelf-wide terrigenous units, and carbonate mounds and platforms under present reefs[48]. Given these limited datasets, the thickness of individual reefs was calculated using the average shelf depth surrounding reef structures, with positive relief above this surface representing reef aggradation across the Pleistocene/Holocene.

Following the above, in both scenarios, we assumed that regions with any reef coverage (fractional area of reef coverage >0; Fig. 6a) had coral reefs that had grown since the LIG. We assigned the total coral reef thickness deposited since the LIG as the modern basement depth (i.e., we assumed the coral reef surface grew to modern sea level) in regions with basement depths shallower than 55 m. Below this bathymetry, we considered that no reef was present during the LIG. To partition coral reef loading across 122 to 0 ka, we made the assumption that the Last Interglacial reef thickness would represent 1.5 times the thickness of Holocene coral reef growth, given the longer time available for LIG reefs to grow with respect to Holocene ones. In our models, we assumed a reef porosity of 40% (that is, the porosity of reefs in sand flats/lagoons in the GBR reported by ref. [65]) and a coral reef density of 1600 kg/m³ (equivalent to the average coral colony density as reported by ref. [66] in ref. [65]).

For the "fine resolution grid" coral loading scenario, we multiplied our map of reef thickness by the fractional area of reef coverage (Fig. 6a). This assumes that the coverage hasn't changed since 120 ka. Accounting for the aerial extent on a fine-resolution grid results in a reduced mass load compared to the "coarse resolution grid" that does not account for a fractional area of reef coverage. The fine resolution grid is characterized by a total volume of $3.1 \times 10^{11}$ m³ (Fig. 4a), whereas the coarse resolution grid's load is greater by an order of magnitude, with a total volume of $5.6 \times 10^{12}$ m³ (Fig. 4c). The last reef loading scenario that accounts for aerial extent by interpolating the fine resolution loading scenario onto the coarser grid (Fig. 4e) results in a substantially smaller total volume ($2.2 \times 10^8$ m³), despite predicting a similar magnitude of relative sea level change compared with that associated with the fine resolution simulation (Fig. 4b, f).

To isolate the impact of reef loading, we did not include ice-sheet loading changes in our modeling. Our reef loading scenario introduced the LIG coral thickness at 120 ka and the Holocene coral thickness at 8 ka. Although coral reefs are built over a longer time span, we simplified our calculation by introducing the load at a single timestep, assuming that the timing of the load will have a negligible impact at present-day after several thousand years of isostatic adjustment. To conserve mass, we uniformly removed a layer of sediment from the continents with a mass equivalent to the total reef load globally.

Although reef loading prior to the LIG would have induced an ongoing isostatic response at the LIG, our analysis is limited to estimating sea-level change since the LIG due to reef loading over only the last glacial cycle. Thus, we limited our modeling to the period from 122–0 ka to assess the magnitude of sea level change due to reef loading since 122 ka.

### Modeling isostatic adjustment: reef isostasy

*1D calculation (coarse resolution)*. To calculate relative sea-level change ($\Delta$RSL) in response to reef loading over the last ice age, we used a gravitationally self-consistent sea-level model. We used the coarse-resolution coral reef loading scenario as input to a 1D sea-level model, which assumes radially symmetric Earth structure. Our calculations are based on the theory and pseudo-spectral algorithm described by Kendall et al.[67] with a spherical harmonic truncation at degree and order 512 (spatial resolution of ~34 km). These calculations include the impact of load-induced Earth rotation changes on sea level[68,69], evolving shorelines, and the migration of grounded, marine-based ice[67,70–72]. Our predictions require models for Earth's viscoelastic structure. We adopted an earth model characterized by a lithospheric thickness of 96 km, and upper and lower mantle viscosities of $5\times10^{20}$ and $5\times10^{21}$ Pa s, respectively, similar to prior models used for Australia[6].

*3D calculation (fine resolution)*. To solve for relative sea level change in response to coral reef loading on a higher resolution of 1 km, we used a global 3D finite volume sea level and Earth deformation model[73]. The numerical approach incorporates lateral variations in Earth's structure and calculates the resulting gravitationally self-consistent sea level change[74]. Previous studies have adopted this computational model in order to account for 3D earth structure (e.g.,[75–77]). The 3D GIA model is capable of km-scale resolution, which is achieved through regional grid refinement for computational efficiency[76]. The importance of fine-resolution GIA modeling has been demonstrated for the solid Earth response to marine grounding line migration in Antarctica[78]. Grid refinement is achieved by incrementally bisecting grid edges in the selected region to achieve the desired $1 \times 1$ km resolution, and a final smoothing operation along the region boundary to ensure a well-behaved transition.

Our simulation uses a 3D viscoelastic earth model. Here, we apply the hybrid model described in Austermann et al.[79], which infers mantle viscosity from seismic tomography using anelastic scaling relationships and additional information on the thermal and rheological state of the upper mantle. In the upper 400 km, a calibrated parameterization of anelastic behavior at seismic frequencies is used to self-consistently determine lithospheric thickness (assumed here to be equivalent to 1175 °C isotherm depth) and viscosity variations from the shear-wave velocity ($V_S$) structure of the tomographic model, SL2013sv[80,81]. Below 400 km, viscosities are derived from the shear-wave tomography model SEMUCB-WM1[82]. Austermann et al.[79] provides details on the $V_S$ to viscosity conversion.

In our 3D GIA calculations, viscosity variations are shifted at each depth to average to $5 \times 10^{20}$ Pa s in the upper mantle viscosity $5 \times 10^{21}$ Pa s in the lower mantle viscosity[6], identical to the earth model used in the 1D GIA calculations. The effective lithospheric thickness in this region varies from 50–100 km. We paired this model with the fine-resolution coral reef loading scenario (Fig. 4a) which accounts for reef coverage area at 1 km resolution (Fig. 6).

### Modeling GIA: ice loading

We modeled relative sea level change in response to the ice sheet and ocean loading changes since the LIG using the 1D pseudo-spectral approach described in Kendall et al.[67]. We used the same model and earth structure described in the 1D reef loading sea-level calculations.

We used an ice history characterized by the GMSL history in Waelbroeck et al.[83] over the last glacial cycle. The ice history was constructed using the ICE-6G deglacial ice geometry history and has no excess melt across the LIG relative to the present day (as in ref.[79]). The GMSL history was adjusted at the LIG since the Waelbroeck GMSL history assumes a value of $-75$ m at 128 ka, which is at odds with coral evidence from the many locations that indicate sea level must have been close to present at that time (see details in ref.[5]). To account for this discrepancy, the timing of the GMSL curve is shifted back prior to the LIG by 3.5 ka. This shift allows for a longer interglacial time period without changing the deglaciation pattern of the original curve and places the MIS 6 sea-level low stand at 135.5 ka (as in ref.[5]).

### Dynamic topography

Observational estimates indicate that mantle flow-driven vertical motions can reach rates of $\sim 0.1$–$1$ m k yr$^{-1}$ in certain locations, suggesting a relevant fraction of relative sea-level change along the Great Barrier Reef from the LIG to present day could result from evolving mantle dynamic topography[10,84–86]. To investigate this possibility, we simulate rates of global dynamic topography change using the mantle convection code ASPECT and an ensemble of Earth models based on 5 seismic tomographic inversions of deep Earth structure (LLNL-G3D-JPS[87]; S40RTS[88]; SAVANI[89]; SEMUCB-WM1[82]; TX2011[90]) and 3 radial viscosity profiles (S10[91]; F10V1[92]; F10V2,[92]).

Above 300 km, input temperature and density fields are determined from seismic velocity using an experimentally derived parameterization of rock anelasticity at seismic frequencies[93]. Uncertain parameters in this formulation are calibrated using a range of independent observational constraints on the co-variation of upper mantle $V_S$, temperature, attenuation, and viscosity (see ref.[80] for details). This approach ensures that the mapping between seismic velocities and buoyancy variations is thermomechanically self-consistent, while also partially correcting for discrepancies between tomographic models that result from parameterization choices rather than true Earth structure. Here, the seismic velocity model we use to obtain upper mantle structure is SLNAAFSA, a version of the SL2013sv upper mantle model[81] into which a number of high-resolution regional updates have been incorporated (see ref.[94] for details). This input structure is chosen since it produces geodynamic predictions that are in good agreement with landscape evolution[95], mantle potential temperature[96], and residual depth observations, even at relatively short wavelengths (~1000 km;[80]).

Below 400 km, a thermodynamic modeling approach is used to obtain thermochemical buoyancy structures for each combination of seismic tomographic and rheological input that are compatible with present-day geophysical observables, including geoid anomalies, dynamic topography, and core-mantle boundary (CMB) excess ellipticity, and comprise thermochemical anomalies within the base of large low-shear-velocity provinces (LLVPs;[97]; see Supplementary Material for further details[37]). Note that, although LLVPs have limited impact on LIG-to-present dynamic topography change, our calculations of the RSL change induced by mantle flow account for associated geoid variations (see Supplementary Material for further details[37]). Since these gravitational changes are more sensitive to the deep mantle, incorporation of accurate LLVP structure in our global convection simulation produces a non-negligible improvement in the reliability of our predictions. Between 300 and 400 km, temperatures and densities derived from these two independent parameterizations are smoothly merged by taking their weighted average as a function of depth.

The time-dependent geodynamic simulations derived from these Earth models assume free-slip conditions at the surface and core-mantle boundary, account for lithospheric cooling by including shallow mantle buoyancy variations and representative thermal conductivity, and incorporate temperature- and composition-dependent viscosity variations (see Supplementary Material for further details[37]). Following[10], we run our models forward in time and, to avoid the potential for transient numerical artefacts in early time steps to affect our results, we assume the average rate of dynamic topography change between 0.5 and 1.5 Ma is representative of that experienced between the LIG and the present day. Change in dynamic topography at specific sea-level sites is calculated by combining perturbations due to the evolving mantle flow pattern with those caused by rigid plate motion across the convective planform. This is accomplished by translating the dynamic topography field calculated for the LIG into its present-day coordinates using plate velocities taken from MORVEL[98], before calculating the difference between this rotated LIG field and the predicted present-day field, yielding a total of 15 individual model predictions (5 tomography models combined with 3 viscosity profiles). Note that the maximum horizontal resolution of the tomographically derived Earth models is ~200 km, placing an important limit on the minimum wavelength of predicted dynamic topography variations.

## Data availability
The data presented in this study, including model outputs, are available open-access (CC-BY 4.0 license) in Zenodo[37], alongside with supplementary text and figures (https://doi.org/10.5281/zenodo.7697073). A preprint of this work (including both pre- and post-review versions is available from EarthArXiv (https://doi.org/10.31223/X55S8X).

## Code availability
ASPECT (version 2.1.0-pre) which was used to perform the mantle convection modeling is available on GitHub[99]. The necessary initial temperature inputs, are also archived alongside example parameter files and dynamic topography predictions in Zenodo[100] (https://doi.org/10.5281/zenodo.8093846).

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

## Acknowledgements

This work was funded by the European Research Council (ERC) under the European Union's Horizon 2020 research and innovation program (grant agreement no. 802414 to A.R.). T.P. acknowledges funding from the NSF EAR Postdoctoral Fellowship, the University of California President's Postdoctoral Fellowship, and NSF OCE—2054757. F.D.R. acknowledges funding from the Imperial College Research Fellowship Scheme. J.A. acknowledges funding from NSF grant OCE-1841888. Funding is also acknowledged by Harvard University (J.X.M. and K.L.). The map in Fig. 1a was created using ArcGIS software by Esri. ArcGIS® and ArcMap™ are the intellectual property of Esri and are used herein under license. Copyright Esri. All rights reserved. For more information about Esri® software, please visit (www.esri.com). We thank the Computational Infrastructure for Geodynamics (geodynamics.org) which is funded by the National Science Foundation under awards EAR-0949446 and EAR-1550901 for supporting the development of ASPECT.

## Author contributions

The parts of the manuscript related to field observations were written by A.R. in collaboration with M. J.O. and I.D.G. The parts of the manuscript related to modeled vertical land motions were written by T.P. in collaboration with F.R., with inputs from J.X.M., J.A., and K.L. The initial concept of this work was developed by A.R., M. J.O., I.D.G., and J.X.M. Models of reef isostasy was developed by T.P. Models of dynamic topography and GIA was developed by F.R., J.A., and K.L.

## Competing interests

The authors declare no competing interests.
