## [Peer Review File · Communications Earth & Environment]

Web links to the author's journal account have been redacted from the decision letters as indicated to maintain confidentiality.

3rd Oct 22

Dear Professor Rovere,

Your manuscript titled "The influence of reef isostasy, dynamic topography, and glacial isostatic adjustment on the Last Interglacial sea-level record of Northeastern Australia." has now been seen by 3 reviewers, and I include their comments at the end of this message. They find your work of interest, but raise some important concerns. We are interested in the possibility of publishing your study in Communications Earth & Environment, but would like to consider your responses to these concerns and assess a revised manuscript before we make a final decision on publication.

Please, take into account these editorial thresholds.

- * Provide a compelling assessment of the relative importance of processes influencing relative sea level in northeastern Australia, including robust dating and calculations
- * Re-balance the results and discussion regarding GIA, dynamic topography and reef isostasy in light of the comments from Reviewer #1
- * Fully explain and justify your methods, approach and assumptions to ensure that your results are clear and reproducible.

We therefore invite you to revise and resubmit your manuscript, along with a point-by-point response that takes into account the points raised. Please highlight all changes in the manuscript text file.

Please use the following link to submit your revised manuscript, point-by-point response to the referees' comments (which should be in a separate document to any cover letter) and the completed checklist:

[link redacted]

We hope to receive your revised paper within six weeks; please let us know if you aren't able to submit it within this time so that we can discuss how best to proceed. If we don't hear from you, and the revision process takes significantly longer, we may close your file. In this event, we will still be happy to reconsider your paper at a later date, as long as nothing similar has been accepted for publication at Communications Earth & Environment or published elsewhere in the meantime.

We understand that due to the current global situation, the time required for revision may be longer than usual. We would appreciate it if you could keep us informed about an estimated timescale for resubmission, to facilitate our planning. Of course, if you are unable to estimate, we are happy to accommodate necessary extensions nevertheless.

Please do not hesitate to contact me if you have any questions or would like to discuss these revisions further. We look forward to seeing the revised manuscript and thank you for the opportunity to review your work.

Best regards,

Maria Laura Balestrieri
External Editor
Communications Earth & Environment

Joe Aslin
Locum Chief Editor
Communications Earth & Environment

EDITORIAL POLICIES AND FORMATTING

Editorial Policy: [Policy requirements](https://www.nature.com/documents/nr-editorial-policy-checklist.pdf) (Download the link to your computer as a PDF.)

Furthermore, please align your manuscript with our format requirements, which are summarized on the following checklist:

[Communications Earth & Environment formatting checklist](https://www.nature.com/documents/commsj-phys-style-formatting-checklist-article.pdf)

and also in our style and formatting guide [Communications Earth & Environment formatting guide](https://www.nature.com/documents/commsj-phys-style-formatting-guide-accept.pdf) .

*** DATA: Communications Earth & Environment endorses the principles of the Enabling FAIR data project (<http://www.copdess.org/enabling-fair-data-project/>). We ask authors to make the data that support their conclusions available in permanent, publically accessible data repositories. (Please contact the editor if you are unable to make your data available).

All Communications Earth & Environment manuscripts must include a section titled "Data Availability" at the end of the Methods section or main text (if no Methods). More information on this policy, is available at <http://www.nature.com/authors/policies/data/data-availability-statements-data-citations.pdf>.

If a community resource is unavailable, data can be submitted to generalist repositories such as [figshare](https://figshare.com/) or [Dryad Digital Repository](http://datadryad.org/). Please provide a unique identifier for the data (for example a DOI or a permanent URL) in the data availability statement, if possible. If the repository does not provide identifiers, we encourage authors to supply the search terms that will return the data. For data that have been obtained from publically available sources, please provide a URL and the specific data product name in the data availability statement. Data with a DOI should be further cited in the methods

reference section.

REVIEWER COMMENTS:

Reviewer #1 (Remarks to the Author):

Review of manuscript "The influence of reef isostasy, dynamic topography, and glacial isostatic adjustment on the Last Interglacial sea-level record of Northeastern Australia.", by a. Rovere et al., September 2022.

Dear authors, dear editor.

This article first identifies a discrepancy between closely located offshore and onshore vertical land motion since the last interglacial stage in NE Australia, and then attempts to explain this observation by modeling the effects of dynamic topography, GIA, and reef isostasy. The latter is proposed as a new concept. I found this article appealing at first glance, but then identified a range of issues at many stages, from observations to modeling on the scientific side, but also on the form of the manuscript itself. Some issues are more critical than others, but I overall feel sorry to write that some problems may even unfortunately be redhibitory. I am sorry for this pessimistic appreciation, and nevertheless wish that my comments will ultimately help.

Regards,
Laurent Husson

Below are my main comments, followed by linear comments.

- I found the database for basic observation under constrained, both onshore and offshore. Regarding offshore sea level indicators, I did not check the references but they seem to be derived from cores (I.3). If so, is there any a priori argument for sample depths being precisely sea level highstands, besides the age with is admittedly close to MIS5e? They could be just before or just after the highstand, in the flesh of a reef unit from MIS5e, not necessarily the highstand but easily a few meters below.

Regarding the onshore data, this is even more of a concern as they are very speculative. While the landforms are admittedly very likely beach ridges, for their morphologies and strandplains are quite clear, their purported LIG age is very speculative and only relies on a single dating from Gagan that is laterally extrapolated alongshore (and OSL ages that are unfortunately "in prep". These additional ages may be correct, and could be used if they were not so critical. But at this stage, I think one can't seriously relying on them). Because the rationale of the study rests on the purported LIG ages of these onshore landforms, I find this to be critical.

- reef isostasy, general comment: There is clearly something that I don't understand here, and it can be my fault. I don't understand the reason why the authors initially inferred that "reef isostasy" could be significant? Could a Turcotte-Schubert-like calculation readily show that it will have a negligible impact? It seems that the authors wanted to make reef isostasy a big thing since the beginning, but I don't understand the reason why, for the load is obviously many many times smaller than any other load known to distort RSL, and in the end, only GIA and dynamic topo matter. In the end, descriptions and computations of reef isostasy overwhelms the manuscript, although it has a very minor impact, and in the end the authors just rule it out (or almost, they ultimately fish it back by arguing I may play a role elsewhere). I understand this was likely frustrating, but one has to admit that this is, in a way, a "non-result".

- reef isostasy, loading: I don't really understand how the reconstructions of MIS5e and Holocene reef loads were carried out. I may have misunderstood, but from what I get, it certainly grossly overestimates the total load, partly by ignoring all constructions prior to MIS5e, by assigning all

the inferred reef sequence to be distributed between Holocene and LIG only. (more details below comments for l. 386)

- reef isostasy, modeling: Why using 3 rather sophisticated methods for such a minor contributor? If it really needs to be presented, why not opting for the most appropriate, and skip the less accurate others?

- reef isostasy, modeling: why using 3D model when working at regional scale, with local loading history only: that seems incompatible with the aim to consider the full spherical, GIA-like resolution; for instance, among other things, the change in Earth's rotation is mentioned: if this is relevant, other contributions from coral reefs elsewhere in the world should also be accounted for, for otherwise the solution is clearly incomplete and meaningless, at least regarding this term of the equation.

- reef isostasy and GIA: if reef isostasy is to be accounted for, why not presenting it as an complementary model to the GIA model: Modeling reef isostasy and GIA are similar, so why not having a regular GIA simulation (with ICE6G but no reef) and another simulation accounting for ICE6G + extra reef loading. And simply showing the difference between the two would quantify the role of reef loading and reveal its minor impact overall.

- dynamic uplift, calculation: it is not very clear how the structure from 0 to 300 km is constructed. Is everything converted into temperature and density anomalies from surface downward? This might not be the best option given the lateral variations in the crust and lithosphere thicknesses in this part of the world. In addition, the response at the degree of interest will be surely dominated by those shallow structures, which are the least constrained. This is already a problem for instantaneous flow solutions, this is even more problematic for time-dependent solutions, as required to compute vertical land motion.

- dynamic uplift, calculation: using global models is often elegant, and the method used here is probably as good as it may be for global models. However, the technical complexity of the models seems to be at odds with the uncertainty of the model outputs (for the above reasons). This is not necessarily a problem, but I found it quite odd (for instance among others the "chemical heterogeneity in the lowermost sections of LLVPs" certainly have no impact on the short wavelengths of uplift and subsidence of interest in the current study).

- I found the text to be difficult to read and to follow the thread. Some parts are redundant, some are not informative enough while not referring to the Methods or SI. I found the content of the main text and the methods unevenly distributed, with an overweight on the impact of isostatic isostasy in the main text (it even has its own section 2), and conversely a very slight description of dynamic uplift and subsidence and GIA (which in the end are the most important here). GIA and dynamic topo respectively have 7 lines and 20 lines in the discussion. The conclusion is that their impact is much more important than reef loading, at odds with the respective space they occupy in the main text.

- Another reason that makes the main text difficult to follow is that it is often implicitly assumed that the reader is already aware of the methods used, and has carefully read the Methods section before reading the main text. Please rewrite to ease the comprehension.

- l. 72: is there any a priori argument for sample depths being precisely sea level highstands, besides the age which is admittedly close to MIS5e? They could be just before or just after the highstand, in the flesh of a reef unit from MIS5e, not necessarily the highstand but easily a few meters below.

- l. 57, 344 and elsewhere: The role of dynamic topo on coral reef subsidence has been advocated before, in this region in particular. I think at least of the work of di Caprio et al from ~2010-2012, and if I may bring up a personal reference, in Husson et al., 2022.

- l. 104: on fig 2b, it seems that the top of the barrier is at about 7.5 m asl, not 6m, why is that?

- l. 111: It does not make sense to go down to the centimeter scale. Even decimeter is probably

- already ambitious...! This is misleading as it gives the erroneous impression of precision.
- l. 136: why MIS7 ? it was never mentioned up to here?
 - l. 198: I don't understand this simulation: unless I misunderstood, it can only be an overestimate, by construction; why even mentioning it?
 - l; 201-205: I understand that they are more localized, but in principle there are also many more point loads. I would think that they will integrate to the same bulk load, and to the same response.
 - l. 305: ie reef isostasy is found negligible in the GBR, this probably holds at all locations in the world... (but I now read in the following lines that it may import "in areas with dense and widespread coral reef coverage". Where if not the GBR?
 - l. 306: why comparing the maximal, local value of reef isostasy to GMSL change? Apples and oranges?
 - l. 307: why not acknowledging that it only has a negligible impact.
 - l. 386: I am not sure that i understand correctly: the assumption is that any morphological structure shallower than 55 m is entirely made up coral reefs, and that this structure was built during LIG and Holocene with a 1.5 ratio? Is there any support for this? I don't know the GBR so well but I believe it is much older than LIG and I anticipate that earlier reef units would have contributed, possibly with a very large share. How is this 55 m depth justified?
- fig. 2: reverse the bottom plot (so that East is on the left).
 - fig. 3 and caption: I have a hard time to understand the reasoning here, because erosion seems to be important whereas only reef loading matters. Could this be clarified in the figure and in the caption to minor the role of erosion (at least for the current reasoning)?
 - fig. 4: left panels are not called upon in the text.
 - fig. 5: axis labels are missing on the ordinates for panels G and H, respectively for reef thickness and elevation change.
 - fig. 5: Why is GIA contribution different between panels B and I-J?

Reviewer #2 (Remarks to the Author):

This is an interesting and novel paper which tackles a long-standing conundrum about why onshore and offshore RSL indicators along the Great Barrier Reef coastline suggest different magnitudes of RSL change during the Last Interglacial. The results will be of interest to workers in this and allied fields. The conclusions are original and are backed up by quantitative analysis and good use of figures to illustrate the main results.

I am not a GIA modeller so cannot comment on the detail of the modelling approaches. I can however comment on the broader aims and scope of the paper and the explanation and use of RSL data. I suggest this paper is suitable for publication with minor corrections.

Comments in order through the text:

Line 9 – spelling of strandplains

Line 1-26 - Why cannot offshore reefs at -5 to -20 m on the GBR be contemporaneous with onshore strandplains at +3 to +9 m without invoking differential vertical land motions? Are the corals known to only live in extremely shallow water? The introduction paragraph needs to make this argument more clearly.

Line 72 -89 - The information on the coral indicators is short and largely refers to the Dechnik et al 2019 paper for information on the LIG corals which provide motivation for this study. It would be helpful to include an extra sentence or two on why these corals cannot indicate RSL above present in the LIG.

Line 90-139 - It would also help to explain why the newly identified LIG strand plains cannot be of Holocene age, given that there was a Holocene highstand and they are undated. It would help to annotate the maps provided as supp info (and Fig 2) with the location of the Holocene equivalents to be clear that they lie onshore and at a higher elevation than the Holocene ones.

Fig 3 suggests that LIG reefs would be subaerially eroded during the glacial phase. Is this taken into account when using them to estimate LIG RSL?

Section 3.1 – should surely be 'fine' vs 'coarse' or 'high' and 'low' rather than 'high' vs 'coarse'.

Line 287 argues that Holocene sediment thickness on the shelf is limited to <2.5 m based on one reference. The authors should consider also referencing the following and potentially other papers on this topic and expanding this section slightly:

Hinestrosa, G., Webster, J. M., and Beaman, R. J. (2016). Postglacial sediment deposition along a mixed carbonate-siliciclastic margin: New constraints from the drowned shelf-edge reefs of the Great Barrier Reef, Australia. *Palaeogeography, Palaeoclimatology, Palaeoecology*, 446:168–185.

McNeil, M., Nothdurft, L., Hua, Q., Webster, J., Moss, P. (2022). Evolution of the inter-reef Halimeda carbonate factory in response to Holocene sea-level and environmental change in the Great Barrier Reef. *Quaternary Science Reviews*, 277(107347), 1-19.

The methods section is largely clear when read alongside the supp info file. Two issues spring to mind though 1) are the 1D and 3D models different in the way that they calculate deformation? If so it pays to be cautious when comparing their results, which could be better emphasised in the text. 2) the explanation of how the reef thickness is calculated is rather brief but very important. The authors should consider having more information on this in the main text.

Reviewer #3 (Remarks to the Author):

This study tackles the long-standing problem of fossil coral reefs of last interglacial age being submerged in the Great Barrier Reef by several tens of metres below modern sea level while on-shore remnants of the former sea-level highstand are preserved at heights above modern sea level. The authors model three potential factors that may cause this discrepancy including reef isostasy (loading of the coral reef), glacio-isostatic adjustments due to water loading on the continental shelf and dynamic topography. The authors conclude that reef isostasy produces a negligible influence of the changes to past sea-level indicators. Changes in the tilt of the continent of NE Australia due to dynamic topography are considered the most likely explanation, although the spatial model resolution cannot fully resolve the observed discrepancies. Overall I find the manuscript to be well-written and makes an important contribution to sea-level research. I believe the paper is suitable for publication in *Nature Communications Earth and Environment* and I have only some minor considerations for the authors for a revision.

Lines 2 and 3: I think where first used the terms LIG and GBR need to be spelt out.

Sentence lines 3-7: the way this is written infers that the emerged strandplain deposits are 3 to 9 m above the LIG coral reef deposits where in fact I think the authors mean that these deposits are 3 to 9 m above modern sea level? Suggest to rewrite to make this clear.

Suggest to add the Ryan et al. (2018) study from Holbourne Island into Figure 1 for completeness as this came out just after the Dechnik study.

I didn't see the supplement section but from what I can gather the yellow markers for the LIG strandplain deposits were from a collection of Gagan et al, Murray-Wallace and Belperio, Goodwin and this current study? Wonder if this is worth clarifying better in here – maybe is there a table in the supplement that can be reference?

Line 135: suggest to delete "records"

Line 277: suggest to change to "While such a mechanism might have relevant local effects..."

Line 338: replace "the" with "that" so will read "...similar to that observed.."

Figure 5: The order in the caption is different to the diagrams (i.e. seems that 5B is the GIA and 5C is the dynamic topography – other way around in the caption).

Line 355 and 393: A couple of instances where LIG is written out – for consistency just go with acronym.

Line 407: suggest to add "are" so will read "...coral reefs are built over..."

Line 402 (also line 480): reference to Figure 6A – I don't see a multi-panel figure so should just be Figure 6?

In the following, we address each of the comments raised in the reviews, and provide a detailed listing of the associated revisions to the text. We intersperse the reviewers' comments (black font) with our responses (in blue font). Line numbers correspond to the version of the revised manuscript with track changes.

Response to Reviewer #1

This article first identifies a discrepancy between closely located offshore and onshore vertical land motion since the last interglacial stage in NE Australia, and then attempts to explain this observation by modeling the effects of dynamic topography, GIA, and reef isostasy. The latter is proposed as a new concept. I found this article appealing at first glance, but then identified a range of issues at many stages, from observations to modeling on the scientific side, but also on the form of the manuscript itself. Some issues are more critical than others, but I overall feel sorry to write that some problems may even unfortunately be redhibitory. I am sorry for this pessimistic appreciation, and nevertheless wish that my comments will ultimately help.

We have attempted to revise the manuscript to address the concerns raised below by the reviewer. Addressing the concerns was relatively straightforward, and doing so has significantly improved the overall value of our MS.

Below are my main comments, followed by linear comments.

- I found the database for basic observation under constrained, both onshore and offshore. Regarding offshore sea level indicators, I did not check the references but they seem to be derived from cores (l.3). If so, is there any a priori argument for sample depths being precisely sea level highstands, besides the age with is admittedly close to MIS5e? They could be just before or just after the highstand, in the flesh of a reef unit from MIS5e, not necessarily the highstand but easily a few meters below. Regarding the onshore data, this is even more of a concern as they are very speculative. While the landforms are admittedly very likely beach ridges, for their morphologies and strandplains are quite clear, their purported LIG age is very speculative and only relies on a single dating from Gagan that is laterally extrapolated alongshore (and OSL ages that are unfortunately "in prep". These additional ages may be correct, and could be used if they were not so critical. But at this stage, I think one can't seriously relying on them). Because the rationale of the study rests on the purported LIG ages of these onshore landforms, I find this to be critical.

We would like to clarify that we did not collect any new field data for this MS. Rather, we relied on published data and interpretations. As far as the beach barriers are concerned, we re-mapped some of the most outstanding Pleistocene barriers, backing the Holocene strandplains, using state of the art Digital Elevation Models from Geoscience Australia. These (and other) strandplains are currently the subject of an extensive study (including dating) by co-author Goodwin, that will include full details on the OSL ages mentioned in the text as being in preparation. We feel that a comprehensive discussion is best left to a different outlet (and a complete paper) to be presented coherently. However, we emphasize that this dataset confirms the correlation of these strandplains to MIS 5e, which is based on the presence, along the New South Wales coasts (south of our study area), of the same geomorphological elements we describe in the manuscript (i.e., a Holocene strandplain backed by a MIS 5e one). At these sites, chronological control is also available via U-series on corals embedded into the beach barrier and Amino Acid Racemization ages, as described in Murray-Wallace and Belperio (1991). Also, a site we included in our compilation (North Stradbroke Island) but did not discuss explicitly in the original manuscript, provides

a further confirmation of our interpretation. To make the reader aware of these details, which in our opinion considerably strengthen the correlation of the beach barriers presented herein with MIS5e, we edited the Section "LIG sea-level indicators" as shown in the track-changes version of the MS (**lines 111-163**).

Concerning the comment on the GBR ages, we note that those ages and interpretations as MIS 5e have been confirmed by different groups (all referenced in the paper), and are widely accepted. Indeed, this has (as discussed in the text) motivated the search for subsidence mechanisms along the GBR (see **lines 186-191** in the track-changes version of the MS). We find no reason to mistrust the previously published data and interpretations.

- reef isostasy, general comment: There is clearly something that I don't understand here, and it can be my fault. I don't understand the reason why the authors initially inferred that "reef isostasy" could be significant? Could a Turcotte-Schubert-like calculation readily show that it will have a negligible impact? It seems that the authors wanted to make reef isostasy a big thing since the beginning, but I don't understand the reason why, for the load is obviously many many times smaller than any other load known to distort RSL, and in the end, only GIA and dynamic topo matter. In the end, descriptions and computations of reef isostasy overwhelms the manuscript, although it has a very minor impact, and in the end the authors just rule it out (or almost, they ultimately fish it back by arguing I may play a role elsewhere). I understand this was likely frustrating, but one has to admit that this is, in a way, a "non-result".

Based on these comments we have restructured the results/discussion to improve the balance between the descriptions of reef isostasy with the other processes that we explore in this study. We remark that, although our study shows that reef isostasy is a minor contributor to vertical displacement in this region over the last glacial cycle, there has been ample and longstanding debate about the role of sediment loading in influencing the elevation of last interglacial sea level markers and in this context an exploration of reef isostasy is warranted. Therefore we view our conclusion that GIA and dynamic topography produce larger signals at is an important result, as is our highlighting of the importance of high resolution modeling for accurately assessing the role of local loading on sea level markers.

- reef isostasy, loading: I don't really understand how the reconstructions of MIS5e and Holocene reef loads were carried out. I may have misunderstood, but from what I get, it certainly grossly overestimates the total load, partly by ignoring all constructions prior to MIS5e, by assigning all the inferred reef sequence to be distributed between Holocene and LIG only. (more details below comments for I. 386)

In the methods section we justify our choice for our modeling time frame:

"Although coral reefs are built over a longer time span, we simplified our calculation by introducing the load at a single timestep, assuming that the timing of the load will have a negligible impact at present-day after several thousand years of isostatic adjustment"

While reef isostasy prior to the last interglacial is unlikely to have a significant impact at present day, It is possible that reef isostasy prior to this time period may have impacted the elevation where interglacial reefs grew. In the methods section we also note that:

"Although reef loading prior to the LIG would have induced an ongoing isostatic response at the LIG, our analysis is limited to estimating sea-level change since the LIG due to reef loading ⁴⁵⁶ over only the last glacial cycle. Thus, we limited our modeling to the period from 122 to 0 ka to assess the magnitude of sea ⁴⁵⁸ level change due to reef loading since 122 ka. "

We are confused regarding the reviewer's point about overestimating the load since we only include the thickness of reefs deposited since the last interglacial, and not before.

- reef isostasy, modeling: Why using 3 rather sophisticated methods for such a minor contributor? If it really needs to be presented, why not opting for the most appropriate, and skip the less accurate others?

- reef isostasy, modeling: why using 3D model when working at regional scale, with local loading history only: that seems incompatible with the aim to consider the full spherical, GIA-like resolution; for instance, among other things, the change in Earth's rotation is mentioned: if this is relevant, other contributions from coral reefs elsewhere in the world should also be accounted for, for otherwise the solution is clearly incomplete and meaningless, at least regarding this term of the equation.

This paper focuses on the local laterally varying sea level signal near the GBR. The rotational signal is not going to produce such a local gradient.

When modeling surface loading in problems such as glacial isostatic adjustment a spherical harmonic degree truncation at degree 512 (resolution of ~34 km) is considered to be sufficient. Our use of multiple models serves to demonstrate that accurate modeling of reef isostasy requires a much higher resolution than is typical in loading calculations - modeling using a coarse resolution leads to a significant overestimation of the sea level change signal due to reef loading. The 3-D model of glacial isostatic adjustment allows us to achieve sufficient resolution to capture the detailed geometry of coral reef loading in the Great Barrier Reef region. We have edited the first paragraph of section 2.1 to better clarify this concept. See **lines 240-252** in the track-changes version of the MS.

In regard to the comment about rotation, the reviewer is obviously correct to argue that the long wavelength sea level variation driven by true polar wander will not be a dominant contributor to such a regional tilting, but we would respond with two points. First, since rotation drives a sea level change it will influence the ocean load in the GBR region which will contribute to a shorter spatial scale signal in sea level. Second, our standard 3D model includes a wide range of processes, including rotational effects. Even if our first point were not important, it would seem strange to use a special, non-rotating version of the code given that including rotation in the sea level solver adds negligible time to the simulation. Using a numerical code for any physical problem does not imply that every physical process modeled in that code is of vital importance.

- reef isostasy and GIA: if reef isostasy is to be accounted for, why not presenting it as an complementary model to the GIA model: Modeling reef isostasy and GIA are similar, so why not having a regular GIA simulation (with ICE6G but no reef) and another simulation accounting for ICE6G + extra reef loading. And simply showing the difference between the two would quantify the role of reef loading and reveal its minor impact overall.

Although we appreciate the reviewer's suggestion, we opted to model these processes separately in order to isolate each of their effects, and this highlights the main point the reviewer is emphasizing above, namely that the signal from reef isostasy is small relative to the GIA signal.

- dynamic uplift, calculation: it is not very clear how the structure from 0 to 300 km is constructed. Is everything converted into temperature and density anomalies from surface downward? This might not be the best option given the lateral variations in the crust and lithosphere thicknesses in this part of the world. In addition, the response at the degree of interest will be surely dominated by those shallow structures, which are the least constrained. This is already a problem for instantaneous flow solutions, this is even more problematic for time-dependent solutions, as required to compute vertical land motion.

We accept that, although the methodology for constructing temperature and density anomalies between 0 and 300 km is described in detail in the cited reference (RHGW20; Richards et al., 2020, JGR), the

description in the main text and supplements should have been more expansive. We have now edited both components to give more methodological detail. In short, we use a suite of independent constraints on the global co-variation of upper mantle V_s , temperature, attenuation, and viscosity to calibrate an anelastic parameterisation (Yamauchi & Takei, 2016, JGR) that allows us to convert seismic velocity models into temperature and density in a thermomechanically self-consistent manner.

Except for the density structure of continental lithosphere, which is largely irrelevant in the context of our time-dependent dynamic topography calculations, we would disagree that shallow structure is the least constrained. Indeed, the tomographically inferred upper mantle temperature and density model applied here has been demonstrated to be consistent with a range of independent constraints on mantle potential temperature, landscape evolution, and present-day geodynamic observables (e.g.; Ball et al., 2021, Nat. Comm.; Stephenson et al., 2021, G-Cubed; Richards et al., 2023, EPSL). Moreover, instantaneous and time-dependent mantle flow simulations that incorporate this buoyancy structure predict dynamic topography that is in particularly good agreement with residual depth anomalies and Pliocene sea-level markers around Australia (Richards et al., 2022, EarthArxiv, doi:10.31223/X5Z652). We are therefore confident that the mantle convection simulations presented here are sufficiently accurate to provide a reasonable estimate of LIG-Recent dynamic topography change across NE Australia.

Specifically, the reviewer is directed toward the revised methods description on **lines 647-675** in the track-changes version of the MS.

Finer methodological detail is also included in the revised supplements.

- dynamic uplift, calculation: using global models is often elegant, and the method used here is probably as good as it may be for global models. However, the technical complexity of the models seems to be at odds with the uncertainty of the model outputs (for the above reasons). This is not necessarily a problem, but I found it quite odd (for instance among others the "chemical heterogeneity in the lowermost sections of LLVPs" certainly have no impact on the short wavelengths of uplift and subsidence of interest in the current study).

We thank the reviewer for their generally positive assessment of our mantle convection modelling strategy. While we accept that, compared to shallow structure, LLVP chemical heterogeneity will have limited impact on LIG-to-recent dynamic topography changes across Australia, predicted dynamic topography-induced RSL change is calculated using the approach of Austermann & Mitrovica (2015, GJI). Since this methodology accounts for variations in geoid height anomalies at each time step—in addition to dynamic topography—global simulations with accurate deep mantle structure (i.e. including LLVP heterogeneity) are required for accurate predictions of RSL change, which refer to changes in the vertical distance between the geoid and crust. We therefore believe the complexity of deep mantle structure used in our models is warranted and will only bolster the accuracy of our calculations.

With the points above in mind, we have now edited the methods to clarify why LLVP structure is relevant to our predictions of LIG-Recent RSL change. See **lines 683-690** in the track-changes version of the MS.

- I found the text to be difficult to read and to follow the thread. Some parts are redundant, some are not informative enough while not referring to the Methods or SI. I found the content of the main text and the methods unevenly distributed, with an overweight on the impact of isostatic isostasy in the main text (it even has its own section 2), and conversely a very slight description of dynamic uplift and subsidence and GIA (which in the end are the most important here). GIA and dynamic topo respectively have 7 lines and 20 lines in the discussion. The conclusion is that their impact is much more important than reef loading, at odds with the respective space they occupy in the main text.

Thank you for this constructive feedback. In response to this comment, and earlier comments by the reviewer, we have rebalanced the text to emphasize all of the processes modeled in our study. Our title includes dynamic topography and glacial isostatic adjustment in addition to reef isostasy, and we agree that the main text should reflect this balance.

- Another reason that makes the main text difficult to follow is that it is often implicitly assumed that the reader is already aware of the methods used, and has carefully read the Methods section before reading the main text. Please rewrite to ease the comprehension.

The methods have been written in accordance with the journal format. However, where relevant, we now make explicit reference to the Methods section to point to where more details can be found.

- l. 72: is there any a priori argument for sample depths being precisely sea level highstands, besides the age which is admittedly close to MIS5e? They could be just before or just after the highstand, in the flesh of a reef unit from MIS5e, not necessarily the highstand but easily a few meters below.

First of all, we do not report sample depths, but the depth of the dated facies as interpreted by Dechnik et al. (2017). This is an important distinction, because while a sample (i.e., a coral) can have a large paleo water depth, the ecological association of corals, coralgall framework and other biotas can indicate quite narrow water depth ranges towards the sea surface. To better convey this concept, we added one column to the supplementary material describing the dataset we used reporting verbatim descriptions from Dechnik et al. (2017).

Another line of reasoning (which we mention here only for the sake of argument, but is not reported in the MS) is that it would not matter much if we admitted that the reef units (or the beach barriers) do not capture exactly the highstand: a few meters of difference in paleo SL forming one or the other would still not explain the large difference in elevation between the onshore and offshore records.

- l. 57, 344 and elsewhere: The role of dynamic topo on coral reef subsidence has been advocated before, in this region in particular. I think at least of the work of di Caprio et al from ~2010-2012, and if I may bring up a personal reference, in Husson et al., 2022.

We have now included reference to these studies in the results and discussion; see **lines 318-320** of the track-changes version of the MS.

- l. 104: on fig 2b, it seems that the top of the barrier is at about 7.5 m asl, not 6m, why is that?

6 m is what is reported in the reference, 7.5m is what we get from the profile we chose to draw. We maintain this is accurate, but we clarified this in the text to avoid confusion. See **lines 139-140** in the track-changes version of the MS

- l. 111: It does not make sense to go down to the centimeter scale. Even decimeter is probably already ambitious...! This is misleading as it gives the erroneous impression of precision.

Done, thanks

- l. 136: why MIS7 ? it was never mentioned up to here?

We agree that this sentence was unclear, and we streamlined it. See **lines 180-184** in the track-changes version of the MS.

- I. 198: I don't understand this simulation: unless I misunderstood, it can only be an overestimate, by construction; why even mentioning it?

As we note above, it is important to include this simulation because such coarse resolution methods are common in loading problems such as a GIA, and the comparison with a fine resolution simulation quantifies the significant level of overestimation one would incur using the former. It is common to consider the thickness of coral reefs as being characteristic of the entire spatial extent of the reef.

- I; 201-205: I understand that they are more localized, but in principle there are also many more point loads. I would think that they will integrate to the same bulk load, and to the same response.

The fine scale load effectively reduces the spatial scale of the loading by introducing pervasive gaps. In order to clarify this point in the manuscript we have added the following text to the results and the method section:

Results section:

"Note that these coarse resolution runs use a 1D GIA model set ²²⁸ up and a loading scenario that does not account for reef coverage area, resulting in a larger volume and mass load for the coarse resolution case (Methods)"

"The discrepancy between fine vs. coarse resolution models is due to the fact that the fine resolution calculation involves a more localized loading geometry (and thus reduced crustal deflection) due to elastic compensation within the lithosphere, compared with the coarse resolution case that overestimates the mass load by not accounting for aerial extent on a finer resolution grid."

Method section:

"For the 'fine resolution grid' coral loading scenario, we multiplied our map of reef thickness by the fractional area of reef coverage (Figure 6A). Accounting for the aerial extent on a fine resolution grid results in a reduced mass load compared to the 'coarse resolution grid' that does not account for fractional area of reef coverage."

- I. 305: ie reef isostasy is found negligible in the GbR, this probably holds at all locations in the world... (but I now read in the following lines that it may import "in areas with dense and widespread coral reef coverage". Where if not the GBR?

Locations that come to mind are reef atolls like Aldabra, in the Indian Ocean, or the Maldives. The latter are composed mostly of Holocene and Pleistocene reef successions on top of a subsiding volcanic edifice, so thanks to catch up mechanisms the Holocene reef growth is possibly thicker than those found on the GBR. The main point is the reef isostasy signal should not be dismissed *a priori*. Moreover, in the paragraph just below this text, we discuss data that should help determine whether reef isostasy can be safely neglected or if it is worth exploring more. See **lines 409-426** in the track changes version of the MS.

- I. 306: why comparing the maximal, local value of reef isostasy to GMSL change? Apples and oranges?

We wanted to give the reader a sense of the level of bias that might be introduced in an analysis of GMSL based on GBR sea level records by neglecting reef isostasy. We have rephrased this text to make this more clear (see **lines 395-398** of the track-changes version of the MS).

- I. 307: why not acknowledging that it only has a negligible impact.

Because it might not be negligible in some locations, see answer above.

- l. 386: I am not sure that I understand correctly: the assumption is that any morphological structure shallower than 55 m is entirely made up of coral reefs, and that this structure was built during the LIG and Holocene with a 1.5 ratio? Is there any support for this? I don't know the GBR so well but I believe it is much older than the LIG and I anticipate that earlier reef units would have contributed, possibly with a very large share. How is this 55 m depth justified?

We admit that this is an estimate, informed by the little data available on LIG + Holocene reef units on the GBR (and elsewhere, for the matter). In general, we surmise that carbonate buildup, i.e., coral reefs have aggraded vertically above the underlying antecedent topography, which based on 2D seismic lines across the Queensland shelf is a simple carbonate ramp morphology. Therefore we make the hypothesis that the height of modern reefal structures can be approximated by the height of the surrounding shelf, and this can be taken as the reef minimum thickness. The value of 55 m is inferred from the average depth to the shelf immediately surrounding the reefs.

To clarify this approach and to give more context, we added one paragraph to this section, citing the most relevant coring work in the region. See **lines 485-499** in the track-changes version of the MS.

- fig. 2: reverse the bottom plot (so that East is on the left).

We accepted the suggestion of the reviewer and modified the figure.

- fig. 3 and caption: I have a hard time to understand the reasoning here, because erosion seems to be important whereas only reef loading matters. Could this be clarified in the figure and in the caption to minor the role of erosion (at least for the current reasoning)?

As we write in the caption, "*in our study we do not model the uplift following reef erosion, which we consider to be balanced with Holocene re-growth.*" Another reasoning behind this choice is that LIG reefs are not completely eroded (as we can still see them today), so the initial subsidence due to reef isostasy is never compensated. Note that we also edited Figure 3 from the previous version, we now show the tilting caused by reef isostasy more clearly.

We hope this clarifies this matter.

- fig. 4: left panels are not called upon in the text.

These panels are now cited in the Methods section. Thank you for catching this!

- fig. 5: axis labels are missing on the ordinates for panels G and H, respectively for reef thickness and elevation change.

These labels are now on the figure.

- fig. 5: Why is GIA contribution different between panels B and I-J?

The GIA contribution is the same, but panel B shows the mean, while the envelope in panel I and J represents the 2-sigma uncertainty as described in the figure caption.

Reviewer #2 (Remarks to the Author):

This is an interesting and novel paper which tackles a long-standing conundrum about why onshore and offshore RSL indicators along the Great Barrier Reef coastline suggest different magnitudes of RSL change during the Last Interglacial. The results will be of interest to workers in this and allied fields. The

conclusions are original and are backed up by quantitative analysis and good use of figures to illustrate the main results.

I am not a GIA modeller so cannot comment on the detail of the modelling approaches. I can however comment on the broader aims and scope of the paper and the explanation and use of RSL data. I suggest this paper is suitable for publication with minor corrections.

Thank you for these positive comments on our study.

Comments in order through the text:

Line 9 – spelling of strandplains

Thank you for catching this, we fixed it.

Line 1-26 - Why cannot offshore reefs at -5 to -20 m on the GBR be contemporaneous with onshore strandplains at +3 to +9 m without invoking differential vertical land motions? Are the corals known to only live in extremely shallow water? The introductory paragraph needs to make this argument more clearly.

The corals retrieved on the GBR are known to live in extremely shallow water, and their shallow paleo water depths were also confirmed by the analysis of coralgal assemblages and sedimentary facies carried out by Dechnik et al. (2017). In the introductory paragraph, we noted that the reefs on the GBR are shallow-water species (**lines 2-6**).

Line 72 -89 - The information on the coral indicators is short and largely refers to the Dechnik et al 2019 paper for information on the LIG corals which provide motivation for this study. It would be helpful to include an extra sentence or two on why these corals cannot indicate RSL above present in the LIG.

We think that this query was addressed in the edit discussed above. Furthermore, to clarify which interpretations were derived from Dechnik et al. (2017), we now include in the supplementary Excel table a column with excerpts from the 2017 paper to clarify the interpreted facies and its associated paleo water depth for each drilled reef.

Line 90-139 - It would also help to explain why the newly identified LIG strand plains cannot be of Holocene age, given that there was a Holocene highstand and they are undated. It would help to annotate the maps provided as supp info (and Fig 2) with the location of the Holocene equivalents to be clear that they lie onshore and at a higher elevation than the Holocene ones.

We have now added the original geological interpretation of Gagan to Figure 2. This was possible due to coring that is not available for the other Holocene - LIG strandplains, therefore we did not modify the supplementary material. However, we believe that the revised Figure 2 may help in the interpretation of the topography of the other barriers.

Fig 3 suggests that LIG reefs would be subaerially eroded during the glacial phase. Is this taken into account when using them to estimate LIG RSL?

We do not take into account erosion, since our estimates of thickness are based on the presently observed thicknesses of coral reef since the last interglacial (which includes any subaerial erosion)

Section 3.1 – should surely be ‘fine’ vs ‘coarse’ or ‘high’ and ‘low’ rather than ‘high’ vs ‘coarse’.

We appreciate this helpful suggestion, and have implemented it in the MS.

Line 287 argues that Holocene sediment thickness on the shelf is limited to <2.5 m based on one reference. The authors should consider also referencing the following and potentially other papers on this topic and expanding this section slightly:

Hinestrosa, G., Webster, J. M., and Beaman, R. J. (2016). Postglacial sediment deposition along a mixed carbonate-siliciclastic margin: New constraints from the drowned shelf-edge reefs of the Great Barrier Reef, Australia. *Palaeogeography, Palaeoclimatology, Palaeoecology*, 446:168–185.

McNeil, M., Nothdurft, L., Hua, Q., Webster, J., Moss, P. (2022). Evolution of the inter-reef Halimeda carbonate factory in response to Holocene sea-level and environmental change in the Great Barrier Reef. *Quaternary Science Reviews*, 277(107347), 1-19.

We appreciate these reference suggestions and have now incorporated them into the text (see **lines 370-379** and **lines 415-417** in the track-changes version of the MS).

The methods section is largely clear when read alongside the supp info file. Two issues spring to mind though 1) are the 1D and 3D models different in the way that they calculate deformation? If so it pays to be cautious when comparing their results, which could be better emphasised in the text. 2) the explanation of how the reef thickness is calculated is rather brief but very important. The authors should consider having more information on this in the main text.

In the main text we have added the following sentence to alert the reader that we use different models to calculate the find resolution and coarse resolution case:

“Note that these coarse resolution runs use a 1D GIA model set up and a loading scenario that does not account for reef coverage area (Methods).”

The 1D GIA model and the 3D GIA model use different methodologies to calculate the sea level equation, however the physics included in these calculations is the same and has been shown to produce almost identical results using the same inputs for ice history and earth structure (Latychev et al., 2005).

The following information is included in the main text about the loading setup:

“Because fine resolution modeling using the 3D sea-level model is computationally expensive, we also tested whether a 1D sea-level model could accurately capture the pattern and magnitude of relative sea level change due to reef isostasy. We used the fine resolution coral reef loading scenario (paired with the 3D sea-level model) and first multiplied the loading grid by the fractional area of reef coverage on a 1 km scale. We then interpolated this loading scenario onto a grid with ~34 km resolution to create a coarse grid that accounts for fractional area of reef coverage (Figure 4E)”

Reviewer #3 (Remarks to the Author):

This study tackles the long-standing problem of fossil coral reefs of last interglacial age being submerged in the Great Barrier Reef by several tens of metres below modern sea level while on-shore remnants of the former sea-level highstand are preserved at heights above modern sea level. The authors model three potential factors that may cause this discrepancy including reef isostasy (loading of the coral reef), glacio-isostatic adjustments due to water loading on the continental shelf and dynamic topography. The authors conclude that reef isostasy produces a negligible influence of the changes to past sea-level indicators. Changes in the tilt of the continent of NE Australia due to dynamic topography are considered the most likely explanation, although the spatial model resolution cannot fully resolve the observed

discrepancies. Overall I find the manuscript to be well-written and makes an important contribution to sea-level research. I believe the paper is suitable for publication in Nature Communications Earth and Environment and I have only some minor considerations for the authors for a revision.

We thank the reviewer for their positive assessment of our study.

Lines 2 and 3: I think where first used the terms LIG and GBR need to be spelt out.

Thank you for pointing this out, and we modified the text accordingly.

Sentence lines 3-7: the way this is written infers that the emerged strandplain deposits are 3 to 9 m above the LIG coral reef deposits where in fact I think the authors mean that these deposits are 3 to 9 m above modern sea level? Suggest to rewrite to make this clear.

Thank you for pointing this out, we modified the sentence to clarify the issue.

Suggest to add the Ryan et al. (2018) study from Holbourne Island into Figure 1 for completeness as this came out just after the Dechnik study.

We thank the reviewer for pointing out the Ryan et al. (2018) work, which was already in our reference list. We chose not to add the site described therein in Figure 1 (and in our analysis) because there is no information on the paleo water depth associated with that reef. In fact, the paleo environment associated with this reef is from descriptions in the Ryan et al. paper (and in Figure 4). However, one point is relevant in this context: the reef described by Ryan et al. is much closer to the shoreline than the others, and is embedded in a different geomorphological setting. We convey this information in a modified version of the second paragraph of the Section “LIG sea-level indicators” (lines 98-110 of the track-changes version of the MS).

I didn't see the supplement section but from what I can gather the yellow markers for the LIG strandplain deposits were from a collection of Gagan et al, Murray-Wallace and Belperio, Goodwin and this current study? Wonder if this is worth clarifying better in here – maybe is there a table in the supplement that can be reference?

The excel spreadsheet referenced in the Supplementary Material now provides the information needed for context on field data.

Line 135: suggest to delete “records”

Accepted

Line 277: suggest to change to “While such a mechanism might have relevant local effects...”

Done, thanks for the suggestion.

Line 338: replace “the” with “that” so will read “...similar to that observed..”

Done, thanks for the suggestion.

Figure 5: The order in the caption is different to the diagrams (i.e. seems that 5B is the GIA and 5C is the dynamic topography – other way around in the caption).

Done, thanks for catching this.

Line 355 and 393: A couple of instances where LIG is written out – for consistency just go with acronym.

Done, thanks for the suggestion.

Line 407: suggest to add “are” so will read “...coral reefs are built over...”

Done, thanks for the suggestion.

Line 402 (also line 480): reference to Figure 6A – I don’t see a multi-panel figure so should just be Figure 6?

Thank you for catching this mistake. We fixed it.

26th Jun 23

Dear Professor Rovere,

Please allow us to sincerely apologise for the long delay in sending a decision on your manuscript titled "The influence of reef isostasy, dynamic topography, and glacial isostatic adjustment on the Last Interglacial sea-level record of Northeastern Australia". It has now been seen again by Reviewers #2 and #3 and by a new Reviewer #4, brought in to replace Reviewer #1 who was unable to provide a second report. The Reviewers' comments appear below. In light of their advice we are delighted to say that we are happy, in principle, to publish a suitably revised version in Communications Earth & Environment under the open access CC BY license (Creative Commons Attribution v4.0 International License).

We therefore invite you to revise your paper one last time to address the remaining concerns of Reviewer #4. At the same time we ask that you edit your manuscript to comply with our format requirements and to maximise the accessibility and therefore the impact of your work.

EDITORIAL REQUESTS:

*****Please take care to match our formatting and policy requirements. We will check revised manuscript and return manuscripts that do not comply. Such requests will lead to delays. *****

SUBMISSION INFORMATION:

OPEN ACCESS:

Communications Earth & Environment is a fully open access journal. Articles are made freely accessible on publication under a [CC BY license](http://creativecommons.org/licenses/by/4.0) (Creative Commons Attribution 4.0 International License). This license allows maximum dissemination and re-use of open access materials and is preferred by many research funding bodies.

For further information about article processing charges, open access funding, and advice and support from Nature Research, please visit <https://www.nature.com/commsenv/article-processing-charges>

At acceptance, you will be provided with instructions for completing this CC BY license on behalf of all authors. This grants us the necessary permissions to publish your paper. Additionally, you will be asked to declare that all required third party permissions have been obtained, and to provide billing information in order to pay the article-processing charge (APC).

[link redacted]

** This url links to your confidential home page and associated information about manuscripts you

may have submitted or be reviewing for us. If you wish to forward this email to co-authors, please delete the link to your homepage first **

Best regards,

Joe Aslin
Senior Editor,
Communications Earth & Environment
<https://www.nature.com/commsenv/>
Twitter: @CommsEarth

Maria Laura Balestrieri
Editorial Board Member
Communications Earth & Environment

REVIEWERS' COMMENTS:

Reviewer #2 (Remarks to the Author):

The authors have addressed all of my previous comments and I now recommend this paper for publication.

Reviewer #3 (Remarks to the Author):

I have read through all the responses from the constructive feedback and am satisfied that the authors have addressed the comments carefully (and particularly the ones I made). I need to state I'm not a modeler and so cannot comment on the specifics of the models used but see that reviewer one has made many valuable constructive comments that appear to have been addressed.

Reviewer #4 (Remarks to the Author):

Review of Rovere et al. "The influence of reef isostasy, dynamic topography, and glacial isostatic adjustment on the Last Interglacial sea-level record of Northeastern Australia."

This paper is a detailed examination of the differences between onshore and offshore sea levels in the LIGs reported for Northeastern Australia. The LIG sea-level records are important for interpreting future sea-level changes, and the paper is worthy of publication in that it tries to improve the accuracy of the sea-level data.

Although I joined the peer review process midway through, I think that the comments from the previous reviewers are reflected. However, some details need to be worked out, such as that some abbreviations, once specified, are not used throughout the manuscript. Including this, I wrote down my comment below. I hope the authors will respond to my comments.

Comments

- Please check the abbreviations, as they may not be used where they should be. (e.g., L61, L68)
- There are some places where there is a space in front of the unit and some places where there is no space, please correct this. (e.g., L4)
- L514–516: The reason for choosing the rheology inputting models is needed. If you refer to previous works, please add the reference.
- L564: "has no excess melt across the LIG". Does "excess melt" mean comparing with the present

condition?

- L568: Please add the reference about the coral evidence showing the discrepancy with the Waelbroeck curve.
- L610-612: "LLVps" and "CMB" are abbreviations, so please be spelt out.

Thank you for the opportunity to review this paper.

Dear Editor, we are grateful for the time you and the reviewers dedicated to improve our MS. We have now responded to all the queries. Below, we provide details (alongside with a track-changes version of the MS). We also attach the Editorial Request table with our comments as a separate file, alongside with MS in latex format, figures and tables.

Response to Reviewer #2

The authors have addressed all of my previous comments and I now recommend this paper for publication.

We would like to thank this reviewer for their constructive comments throughout the review process.

Response to Reviewer #3

I have read through all the responses from the constructive feedback and am satisfied that the authors have addressed the comments carefully (and particularly the ones I made). I need to state I'm not a modeler and so cannot comment on the specifics of the models used but see that reviewer one has made many valuable constructive comments that appear to have been addressed.

We would like to thank this reviewer for their constructive comments throughout the review process.

Response to Reviewer #4

This paper is a detailed examination of the differences between onshore and offshore sea levels in the LIGs reported for Northeastern Australia. The LIG sea-level records are important for interpreting future sea-level changes, and the paper is worthy of publication in that it tries to improve the accuracy of the sea-level data. Although I joined the peer review process midway through, I think that the comments from the previous reviewers are reflected. However, some details need to be worked out, such as that some abbreviations, once specified, are not used throughout the manuscript. Including this, I wrote down my comment below. I hope the authors will respond to my comments.

We would like to thank this reviewer for their comments, which helped bring the MS to a final stage. We are answering point-by-point below.

Comments

- Please check the abbreviations, as they may not be used where they should be. (e.g., L61, L68)
Done, we went through the text trying to spot all instances where abbreviations were not used.

- There are some places where there is a space in front of the unit and some places where there is no space, please correct this. (e.g., L4)

Done, we went through the text trying to spot all instances where there was no space in front of the unit and added it.

- L514–516: The reason for choosing the rheology inputting models is needed. If you refer to previous works, please add the reference.

At the end of this sentence we added "similar to prior models used for Australia (reference 71)

- L564: "has no excess melt across the LIG". Does "excess melt" mean comparing with the present condition?

At the end of this sentence, we added "relative to present day"

- L568: Please add the reference about the coral evidence showing the discrepancy with the Waelbroeck curve.

We added a reference to Dyer et al., where this discrepancy is discussed more in detail (we use the same models they employed)

- L610-612: "LLVps" and "CMB" are abbreviations, so please be spelt out.

These stand for "large low-shear-velocity provinces". CMB is "core mantle boundary". We defined these abbreviations in parentheses the first time they are brought up